



# Exploration of the atmospheric chemistry of nitrous acid in a coastal city of southeastern China: Results from measurements across four seasons

**Baoye Hu[1,2,3], Jun Duan[4], Youwei Hong[1,2], Lingling Xu[1,2], Mengren Li[1,2], Yahui Bian[1,2], Min Qin[4*], Wu Fang[4], Pinhua Xie[1,3,4,5], Jinsheng Chen[1,2*]**

[1]Center for Excellence in Regional Atmospheric Environment, Institute of Urban Environment, Chinese Academy of Sciences, Xiamen 361021, China

[2]Key Lab of Urban Environment and Health, Institute of Urban Environment, Chinese Academy of Sciences, Xiamen 361021,
China

[3]University of Chinese Academy of Sciences, Beijing 100086, China

[4]Key Laboratory of Environment Optics and Technology, Anhui Institute of Optics and Fine Mechanics, Chinese Academy of Sciences, Hefei, 230031, China

[5]School of Environmental Science and Optoelectronic Technology, University of Science and Technology of China, Hefei,
230026, China

*Correspondence to*: Jinsheng Chen (jschen@iue.ac.cn) &Min Qin (mqin@aiofm.ac.cn)

**Abstract.** Because nitrous acid (HONO) photolysis is a key source of hydroxyl (OH) radicals, identifying the atmospheric
sources of HONO is essential to enhance the understanding of atmospheric chemistry processes and improve the accuracy of simulation models. We performed seasonal field observations of HONO in a coastal city of southeastern China, along with measurements of trace gases, aerosol compositions, photolysis rate constants ($J$), and meteorological parameters. The results showed that the average observed concentration of HONO was $0.54 \pm 0.47$ ppb. Vehicle exhaust emissions contributed an average of 1.64% to HONO, higher than the values found in most other studies, suggesting an influence from diesel vehicle
emissions. The mean conversion frequency of $NO_2$ to HONO in the nighttime was the highest in summer due to water droplets was evaporated under the condition of high temperatures. Based on a budget analysis, the rate of emission from unknown sources ($R_{unknown}$) was highest at midday, with values of 14.78 ppb·h$^{-1}$ in summer, 6.49 ppb·h$^{-1}$ in autumn, and 2.18 ppb·h$^{-1}$ in spring. Unknown sources made up the largest proportion of all sources in summer (84.92%), autumn (80.29%), and spring (49.98%), whereas the main source in winter was the homogeneous reaction of NO with OH (56.15%), due to winter having
the highest NO concentration of the four seasons. The value of $R_{unknown}$ had a positive logarithmic relationship with the photolysis of particulate nitrate in spring, summer, and autumn. However, $R_{unknown}$ was limited by particulate acidity under the



condition of photolysis of particulate nitrate ($J$ ($NO_3^-$_R) $\times$ $pNO_3^-$) $> 1\ \mu g \cdot m^{-3} \cdot s^{-1}$ in autumn and $J(NO_3^-$_R$) \times pNO_3^- > 2\ \mu g \cdot m^{-3} \cdot s^{-1}$ in spring and summer. The variation of HONO at night can be exactly simulated based on the HONO/$NO_x$ ratio, while the main sources should be considered for daytime simulations. Compared with $O_3$ photolysis, HONO

photolysis has long been an important source of OH, particularly in the morning in spring and winter and around noon in summer and autumn. This study draws a full picture of the sources of HONO across all four seasons and improves the comprehension of HONO chemistry in southeastern coastal China.

## 1 Introduction

Nitrous acid (HONO) photolysis produces hydroxyl (OH) radicals, an important oxidant, in the troposphere (Zhou et al., 2011).
Hydroxyl radicals play an important role in triggering the oxidation of volatile organic compounds and therefore determine the fate of many anthropogenic atmospheric pollutants (Lei et al., 2018). Recent research results have shown that HONO production is the cause of an increase in secondary pollutants (Li et al., 2010;Gil et al., 2019;Fu et al., 2019). Though extensive studies have been conducted in the four decades since the first clear measurement of HONO (Perner and Platt, 1979), the HONO formation mechanisms are still elusive, especially during the daytime, when there is a large difference between
measured concentrations and those calculated from known gas-phase chemistry (Sörgel et al., 2011). Identification of the sources of atmospheric HONO and exploration of its formation mechanisms are beneficial for enhancing our comprehension of atmospheric chemistry processes and improving the accuracy of atmospheric simulation models.

Commonly accepted HONO sources include direct emission from motor vehicles (Chang et al., 2016;Kirchstetter et al., 1996;Kramer et al., 2020;Xu et al., 2015) or soil (Su et al., 2011;Tang et al., 2019;Oswald et al., 2013), the homogeneous
conversion of NO by OH (Seinfeld and Pandis, 1998;Kleffmann, 2007), and the heterogeneous reaction of $NO_2$ on humid surfaces (Alicke, 2002;Finlayson-Pitts et al., 2003). Other heterogeneous daytime sources, such as photosensitive reduction of $NO_2$ on organic surfaces (Stemmler et al., 2006) and the photolysis of particulate nitrate by ultraviolet (UV) light (Kasibhatla et al., 2018;Romer et al., 2018;Ye et al., 2017;Scharko et al., 2014), have been identified by previous laboratory measurements and field studies. Most previous field studies have shown an absence of major HONO sources during the daytime, which is an
important area for further study. According to an analysis of 15 sets of field observations around the world (Elshorbany et al., 2012), the HONO/$NO_x$ ratio (0.02) predicts well HONO concentrations under different atmospheric conditions. To avoid the problem of underestimation, in this study, an empirical parameterization was applied to estimating the HONO concentration, because the current understanding of HONO formation mechanisms is incomplete.

Field measurements of HONO and its precursor $NO_2$ at sites with different aerosol load & composition, and relative humidity
(RH) are necessary to deepen our knowledge of the HONO formation mechanisms. Such measurements have been carried out in coastal cities in China, including Guangzhou (Qin et al., 2009), Hong Kong (Xu et al., 2015), and Shanghai (Cui et al., 2018), where the air pollution is relatively severe (Wang et al., 2017b). However, there has been a lack of research into HONO

in coastal cities with good air quality, low concentrations of NO$_x$ and PM$_{2.5}$, but strong sunlight and high humidity. Insufficient research on coastal cities with good air quality has resulted in certain obstacles to assessing the photochemical processes in

these areas. Due to different emission-source intensities and ground surfaces, the atmospheric chemistry of HONO in the southeastern coastal area of China is predicted to have different pollution characteristics from those found in other coastal cities. Furthermore, HONO contributes to the atmospheric photochemistry differently depending on the season (Li et al., 2010). Therefore, observations of atmospheric HONO across different seasons in the southeastern coastal area of China are urgently needed.

Incoherent broadband cavity-enhanced absorption spectroscopy (IBBCEAS) was employed in this study to determine HONO concentrations in the southeastern coastal city of Xiamen in August (summer), October (autumn), and December (winter) 2018 and March (spring) 2019. In addition, a series of other relevant trace gases, meteorological parameters, and photolysis rate constants were measured at the same time to provide supplementary information to reveal the HONO formation mechanisms. The main purposes of this study were to (1) quantify the gas-phase photostationary state of HONO, (2) calculate the values of

unknown HONO daytime sources, (3) analyze the processes leading to HONO formation, (4) simulate HONO concentrations based on an empirical parameterization, and (5) evaluate OH production from HONO from 07:00 to 16:00 local time. All of these results were compared between the seasons.

## 2 Methodology

### 2.1 Site description

Our field observations were carried out ~80 m above the ground at a supersite located on the top of the Administrative Building of the Institute of Urban Environment (IUE), Chinese Academy of Sciences (24.61° N, 118.06° E) in Xiamen, China in August, October, and December 2018, and March 2019 (Fig. 1). The supersite was equipped with a complete set of measurement tools, including those for measuring gas and aerosol species composition, meteorology parameters, and photolysis rate constants, which provided a good chance to study the atmospheric chemistry of HONO in a coastal city of southeastern China.

### 2.2 Instrumentation

The atmospheric concentrations of both HONO and NO$_2$ were determined using IBBCEAS, which has previously been widely applied to such measurements (Tang et al., 2019;Duan et al., 2018;Min et al., 2016). Multiple reflections in the resonator cavity enhance the length of the effective absorption path, thereby enhancing the detection sensitivity of the instrument. The 1σ detection limits for HONO and NO$_2$ were 60 ppt and 100 ppt, respectively, and the time resolution was 1 min. The

measurement error for HONO and NO$_2$ was estimated to be about 9%. A specific description of the structure and principle of IBBCEAS can be found in a previous report (Duan et al., 2018).



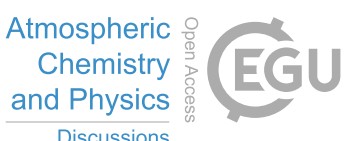

The inorganic composition of $PM_{2.5}$ aerosols, including $Cl^-$, $NO_3^-$, $SO_4^{2-}$, $NH_4^+$, $Na^+$, $K^+$, $Ca^{2+}$, and $Mg^{2+}$, were determined using a Monitor for AeRosols and Gases in ambient Air (MARGA, Model ADI 2080, Applikon Analytical B.V., the Netherlands) with a temporal resolution of 1 h. The MARGA utilizes a steam-jet aerosol collector (SJAC), and online ion chromatography was applied to determine the aqueous sample streams produced by the SJAC. Specific descriptions of the SJAC can be found in previous reports (Slanina et al., 2001;Wyers et al., 1993).

Photolysis frequencies were determined using a photolysis spectrometer (PFS-100, Focused Photonics Inc., Hangzhou, China). These were calculated by multiplying the actinic flux $F$, quantum yield $\varphi(\lambda)$ and the known absorption cross section $\sigma(\varphi)$. The measurements included the photolysis rate constants $J$ ($O^1D$), $J$ (HCHO_M), $J$ (HCHO_R), $J$ ($NO_2$), $J$ ($H_2O_2$), $J$ (HONO), $J$ ($NO_3$_M) and $J$ ($NO_3$_R), and the spectral band ranged from 270 to 790 nm. Hemispherical ($2\pi$ sr) angular response deviations were within ±5%.

The $O_3$ concentration was determined by UV photometric analysis [Model 49$i$, Thermo Environmental Instruments (TEI) Inc.], and the detection limit of the TEI Model 49$i$ is 1.0 ppb. The NO concentration was determined by a chemiluminescence analyzer (TEI model 42$i$) with a molybdenum converter, and the detection limit of the TEI model 42$i$ is 0.5 ppb. Although the TEI model 42$i$ also measures the concentration of $NO_2$, this value might actually include other active nitrogen components. Therefore, the $NO_2$ concentration as measured by IBBCEAS was used in this study. An oscillating microbalance with a tapered element was applied to determine the $PM_{2.5}$ concentration. Meteorological parameters were determined by an ultrasonic atmospherium (150WX, Airmar, USA). The time resolution of all instruments was unified to 1 h to facilitate comparison.

## 3 Results and discussion

### 3.1 Overview of data

The average measured ambient HONO concentration at the measurement site for all measurement periods was $0.54 \pm 0.47$ ppb. The maximum value (3.51 ppb) appeared at 08:00 on 4 December 2018. The HONO mixing level in Xiamen was close to the values found in Rome (0.58 ppb), Nanjing (0.69 ppb), and Hong Kong (0.72 ppb), but was much lower than those in Xi'an (1.04 ppb), Kathmandu (1.05 ppb), Jinan (1.14 ppb), Santiago (2.25 ppb), or Guangzhou (2.75 ppb), as shown in Table 1. Table 1 also shows the seasonal patterns of HONO and related parameters during the night and the day.

In the daytime (06:00–18:00, including 06:00, local time (LT)), the highest concentration of HONO was found in spring and summer (0.72 ppb), followed by winter (0.61 ppb) and autumn (0.50 ppb). In short, the seasonal variation of HONO was well correlated with the seasonality of RH, with high RH in spring (83.31%) and summer (84.58%), followed by winter (75.79%) and autumn (66.47%). In conditions of low RH, the adsorption rate of $NO_2$ is not as rapid as that of HONO, resulting in a reduction in the conversion rate of $NO_2$ to HONO and thus a reduction in the concentration of HONO (Stutz et al., 2004). This seasonal variation in HONO concentration was different from those measured in Jinan (Li et al., 2018), Nanjing (Liu et al.,



2019b), and Hong Kong (Xu et al., 2015). The elevated HONO concentrations in summer, when there is strong solar radiation, suggests the existence of strong sources of HONO and its important contribution to the production of OH radicals. Interestingly, the HONO concentration in the nighttime was lower than that in the daytime in all four seasons. Most previous studies have

found that the HONO concentration at night is significantly higher than that during the day (Wang et al., 2015;Liu et al., 2019c;Li et al., 2018;Elshorbany et al., 2009;Acker et al., 2006;Yu et al., 2009). Coastal cities are susceptible to sea and land breezes, with sea breezes blowing during the day and land breezes blowing during the night (Wagner et al., 2012). Therefore, the concentration of sea salt, as calculated based a previous report (Liu et al., 2020), is significantly higher during the day than that during the night ($P < 0.05$). It is possible that significantly more HONO could be produced by photolysis of sea salts

against the daytime photolysis of HONO (Kasibhatla et al., 2018). Similar results were found in Hong Kong, which is also a coastal city, which further validates the rationality of this assumption (Xu et al., 2015).

The ratio of HONO to $NO_x$ or the ratio of HONO to $NO_2$ have been extensively applied to indicate heterogeneous conversion of $NO_2$ to HONO (Li et al., 2012;Liu et al., 2019c;Zheng et al., 2020). Compared with the $HONO/NO_2$ ratio, the $HONO/NO_x$ ratio can better avoid the influence of primary emissions (Liu et al., 2019c). In this study, the $HONO/NO_x$ ratios during the

day were higher than those during the night, indicating that light promotes the conversion of $NO_x$ to HONO. The highest daytime $HONO/NO_x$ ratio was found in summer (0.072), followed in turn by autumn (0.048), spring (0.034), and winter (0.023). The elevated $HONO/NO_x$ ratio in summer indicates a greater net HONO production (Xu et al., 2015). The low $HONO/NO_x$ ratio in winter can probably be ascribed to heavy emissions and high concentrations of NO in winter (Table 1). The $HONO/NO_x$ ratios during every season in Xiamen were in general higher than those found in studies of other cities, which indicates greater

net HONO production in Xiamen.

The diurnal patterns of HONO, $NO_x$, $HONO/NO_x$, and $J(NO_2)$ averaged for every hour in each season are shown in Fig. 2. As shown in Fig. 2a, the HONO concentration had similar diurnal variation patterns across the four seasons. The maximum values of the HONO concentration were 1.12 ppb in winter, 1.03 ppb in summer, 0.98 ppb in spring, and 0.65 ppb in autumn, and these occurred in the morning rush hour (07:00–08:00), which indicates that direct vehicle emissions may be a significant

source of HONO. The contribution of direct vehicle emissions to HONO will be quantified in Sect. 3.2. The HONO concentration reduced rapidly from the morning rush hour to sunset, and this was caused by rapid photolysis combined with increased height of the boundary layer. The minimum values of HONO concentration were 0.47 ppb in spring, 0.23 ppb in winter, 0.21 ppb in summer, and 0.14 ppb in autumn, and these appeared at sunset, between 16:00 and 18:00. The HONO concentration increased gradually after sunset, which indicates that release from HONO sources exceeded its dry deposition

(Wang et al., 2017a). There was a slight difference in the diurnal variation of HONO between autumn and the other seasons. A rapid reduction of HONO after the morning rush hour was found in spring, summer, and winter. In comparison, the HONO in autumn had an almost constant concentration between 07:00 and 11:00 because $NO_x$ decreased slowly during this period.

As shown in Fig. 2b, $NO_x$ concentration followed an expected profile in the four seasons, with peaks of 45.58 ppb in winter, 40.47 ppb in spring, 32.47 ppb in summer, and 20.07 ppb in autumn, each occurring in the morning rush hour at 10:00, 09:00,





08:00, and 07:00 local time, respectively. After these peaks, $NO_x$ decreased during the day in each season, probably due to photochemical transformation and increasing boundary-layer depth. The $NO_x$ concentrations then began to rise from their minima of 8.20 ppb in summer, 8.85 ppb in autumn, 18.10 ppb in winter, and 23.09 ppb in spring after 14:00, 13:00, 15:00, and 16:00 local time, respectively, which was caused by a combination of weak photochemical transformation and reduction in the boundary-layer depth. The $NO_x$ concentrations during winter and spring were significantly higher than those during autumn and summer. Both the maxima and minima of $NO_x$ appeared later in spring and winter compared with summer and autumn.

It is possible to better describe the behavior of HONO using the HONO/$NO_x$ ratio. The higher HONO/$NO_x$ ratio found at noon in the different seasons, especially in summer and autumn (Fig. 2c), indicates an unknown daytime HONO source. It is worth noting that the maximum value of this ratio in summer (0.147) was significantly higher than the maximum in other seasons, especially in winter (0.034). Fig. 2d shows that the value of the HONO/$NO_x$ ratio increased with the photolysis of $NO_2$ in summer and autumn, suggesting that the unknown HONO source is probably correlated with light (Xu et al., 2015;Wang et al., 2017a;Li et al., 2018;Li et al., 2012). The increase in the HONO/$NO_2$ ratio during the day can be seen more clearly in Fig. 3, and its high value indicates a high HONO production efficiency, which cannot be ascribed to $NO_2$ conversion due to the weak correspondence between HONO and $NO_2$ in three of the seasons (excluding winter). Furthermore, high HONO/$NO_2$ ratios were accompanied by high $J(NO_2)$ in summer, which indicates that HONO formation during the daytime is controlled by light rather than Reaction (R1).

$$NO_2 + NO_2 + H_2O \xrightarrow{\text{surf}} HONO + HNO_3 \hspace{2cm} \text{(R1)}$$

However, the observed maxima can also be ascribed to sources independent from $NO_x$ concentration, such as soil emissions (Su et al., 2011) and photolysis of particulate nitrate (Zhou et al., 2011;Ye et al., 2016), which are not influenced by the decrease of $NO_x$ concentration around noon. A more specific discussion of daytime HONO sources considering the photolysis of particulate nitrate will be given in Sect. 3.4.3. Although the solar radiation intensity in spring and winter was nearly equal, the difference in the HONO/$NO_x$ ratios in these seasons was large, indicating that the solar radiation intensity was not the only factor determining the HONO/$NO_x$ ratio. The HONO emissions from soil were estimated to be 2–5 ppb h$^{-1}$ (Su et al., 2011). However, soil emission was a negligible source of HONO in this study since the surrounding soil is not used for agriculture, and this greatly reduces the amount of HONO released due to no fertilization process (Su et al., 2011).

**3.2 Direct vehicle emission of HONO**

The consistent diurnal variations in HONO and $NO_x$ presented in Sect. 3.1 (Fig. 2) also indicate HONO emissions from local traffic. Five criteria were applied to choose cases that guaranteed the presence of fresh plumes (Xu et al., 2015;Liu et al., 2019c): (1) UV < 10 W·m$^{-2}$; (2) short-duration air masses (<2 h); (3) HONO correlating well with $NO_x$ ($R^2 > 0.60$, $P < 0.05$);




(4) $NO_x > 20$ ppb (highest 25% of $NO_x$ value); and (5) $NO/NO_x > 0.50$. A total of 34 cases met these strict criteria for estimation of the HONO vehicle emission ratios. The slopes of scatter plots of HONO vs $NO_x$ were used as the emission factors.

A total of 34 vehicle emission plumes are summarized in Table 2, and these were used for estimation of the vehicle emission ratios. The plumes were considered to be truly fresh because the mean $\Delta NO/\Delta NO_x$ ratio of the selected air masses was 92%. Vehicle plumes unavoidably mixing with other air masses resulted in the correlation coefficients ($R^2$) between HONO and
$NO_x$ varying among the cases, and these ranged from 0.61 to 0.92. The obtained $\Delta HONO/\Delta NO_x$ ratios ranged from 0.24% to 4.76%, with an average value (±SD) of (1.64 ± 0.95) %. These $\Delta HONO/\Delta NO_x$ ratios have comparability to those obtained in Guangzhou (1.4% (Qin et al., 2009); 1.8% (Li et al., 2012)) and Houston (1.7% (Rappenglück et al., 2013)), but are significantly higher than those measured in Jinan (0.53% (Li et al., 2018)) and Santiago (0.8% (Elshorbany et al., 2009)). The types of vehicle engine, the use of catalytic converters, and different fuels will affect the vehicle emission factors (Kurtenbacha
et al., 2001). A potential reason for the relatively higher $\Delta HONO/\Delta NO_x$ values in our study is that heavy-duty diesel vehicles pass by on the surrounding highway (Rappenglück et al., 2013). It is necessary to examine the specific vehicle emission factors in target cities because of these differences in $\Delta HONO/\Delta NO_x$ ratios. Roughly assuming that $NO_x$ mainly arises from vehicle emissions, a mean $\Delta HONO/\Delta NO_x$ value of 1.64% was used as the emission factor in this study, and this value was adopted to estimate the contribution of vehicle emissions $P_{emis}$ to the HONO concentration using

$$P_{emis} = NO_x \times 0.0164. \tag{1}$$

We can then obtain the corrected HONO concentration ($HONO_{corr}$) for further analysis from the equation

$$HONO_{corr} = HONO - P_{emis}. \tag{2}$$

### 3.3 Nighttime heterogeneous conversion of NO₂ to HONO

### 3.3.1 Conversion rate of NO₂ to HONO

Nighttime $HONO_{corr}$ concentrations can be estimated from the heterogeneous conversion reaction (Meusel et al., 2016;Alicke, 2002;Su et al., 2008a). Although the mechanism of the nighttime HONO heterogeneous reaction is unclear, the formula for the heterogeneous conversion ($C_{HONO}^0$) of NO₂ to HONO can be expressed as

$$C_{HONO}^0 = \frac{[HONO_{corr}]_{t_2} - [HONO_{corr}]_{t_1}}{(t_2 - t_1) \times \overline{[NO_2]}}, \tag{3}$$

where $\overline{[NO_2]}$ is the mean value of NO₂ concentration between $t_1$ and $t_2$. Eq. (4) has been suggested as a way to avoid the
interference of direct emissions and diffusion (Su et al., 2008a):

$$C_{HONO}^X = \frac{\left(\frac{[HONO_{corr}]_{(t_2)}}{[X]_{t_2}} - \frac{[HONO_{corr}]_{(t_1)}}{[X]_{(t_1)}}\right)\overline{[X]}}{(t_2-t_1)\frac{1}{2}\left(\frac{[NO_2]_{(t_2)}}{[X]_{(t_2)}} + \frac{[NO_2]_{(t_1)}}{[X]_{(t_1)}}\right)\overline{[X]}} = \frac{2\left(\frac{[HONO_{corr}]_{(t_2)}}{[X]_{t_2}} - \frac{[HONO_{corr}]_{(t_1)}}{[X]_{(t_1)}}\right)}{(t_2-t_1)\left(\frac{[NO_2]_{(t_2)}}{[X]_{(t_2)}} + \frac{[NO_2]_{(t_1)}}{[X]_{(t_1)}}\right)}, \tag{4}$$

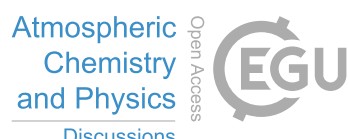

where $[HONO_{corr}]_t$, $[NO_2]_t$, and $[X]_t$ were the concentrations of HONO, $NO_2$, and species used for normalization (including $NO_2$, CO, and black carbon (BC) in this study), respectively, at time $t$, $\overline{X}$ is the average concentration of reference species between $t_1$ and $t_2$, and $C_{HONO}^X$ represents the conversion rate normalized against reference species $X$ (Su et al., 2008a). There

were 91 cases meeting the criteria. Such a large number of cases contributes to the statistical analysis of the heterogeneity of HONO formation. The average values of $C_{HONO}^0$, $C_{HONO}^{NO_2}$, $C_{HONO}^{CO}$, and $C_{HONO}^{BC}$ were $0.48\%\,h^{-1}$, $0.46\%\,h^{-1}$, $0.47\%\,h^{-1}$, and $0.46\%\,h^{-1}$, respectively. The combined $C_{HONO}^C$ was $0.47\%\,h^{-1}$. The average $C_{HONO}$ values obtained using different normalization methods agreed well. Therefore, an estimation value of $0.47\%\,h^{-1}$ should be suitable for the nighttime conversion rate from $NO_2$ to HONO.

We also compared the conversion rates calculated in this study with other experiments. As shown in Table 3, $C_{HONO}^C$ varied widely, from $0.29\%\,h^{-1}$ to $2.40\%\,h^{-1}$, which may be due to the various kinds of land surface in the various environments. The $C_{HONO}^C$ in Xiamen is comparable to those derived in Shanghai ($0.70\%\,h^{-1}$ (Wang et al., 2013)), Jinan ($0.68\%\,h^{-1}$ (Li et al., 2018)), and Hong Kong ($0.52\%\,h^{-1}$ (Xu et al., 2015)), less than the values calculated from most other sites, including Xinken ($1.60\%\,h^{-1}$ (Su et al., 2008a)), Guangzhou ($2.40$ (Li et al., 2012)), Spain ($1.50$ (Sörgel et al., 2011)), Beijing ($0.80$ (Wang et

al., 2017a)), the eastern Bohai Sea ($1.80\%\,h^{-1}$ (Wen et al., 2019)), and Kathmandu ($1.40\%\,h^{-1}$ (Yu et al., 2009)), but more than the value obtained in Shandong ($0.29\%\,h^{-1}$ (Wang et al., 2015)). The highest $C_{HONO}^C$ was found in summer, with a value of $0.55\%\,h^{-1}$, which will be explained in Sect. 3.3.2. Another study also found that the highest $C_{HONO}^C$ ($1.00\%\,h^{-1}$) appeared in summer (Wang et al., 2017a).

### 3.3.2 The influence of relative humidity on HONO formation

The hydrolysis of $NO_2$ on wet surfaces producing HONO is first-order affected by the concentration of $NO_2$ (Finlayson-Pitts et al., 2003;Jenkin et al., 1988) and the absorption of water on the surfaces (Finlayson-Pitts et al., 2003;Kleffmann et al., 1998). A scatter plot of $HONO_{corr}/NO_2$ vs RH is shown in Fig. 4. We calculated the top-five $HONO_{corr}/NO_2$ ratios in every 5% RH interval based on a method introduced in previous literature (Li et al., 2012;Stutz et al., 2004), which will reduce the influence of those circumstances such as advection, the time of the night, and the surface density. These averaged maxima and standard

deviations are shown in Fig. 4 as orange squares, except where data were sparse in a particular 5% RH interval.

As for autumn and winter, the influence of RH on $HONO_{corr}/NO_2$ can be divided into two parts. The RH promoted an increase in $HONO_{corr}/NO_2$ for RH values less than 77.96% in autumn and 91.99% in winter, which is in line with the reaction kinetics of Reaction (R1). However, RH inhibits the conversion of $NO_2$ to HONO when RH is higher than a turning point. According to many previous studies, water droplets will be formed on the surface of the ground or of aerosols when RH exceeds a certain

value, thus resulting in a negative dependence of $HONO_{corr}/NO_2$ on RH (He et al., 2006;Zhou et al., 2007). A similar phenomenon was also found in Guangzhou and in Shanghai (70%, (Li et al., 2012;Wang et al., 2013)) and in Kathmandu and in Beijing (65%, (Yu et al., 2009;Wang et al., 2017a)). However, in summer, RH appeared to promote the increase of





$HONO_{corr}/NO_2$ without a turning point, suggesting that HONO production at night in summer strongly depends on RH. Another study also found a similar phenomenon in the summer in Guangzhou (Qin et al., 2009). This phenomenon might be caused by water droplets being destroyed by high temperatures. This is the reason for the highest $C_{HONO}^C$ in summer. As for spring, the relationship between $HONO_{corr}/NO_2$ and RH is very complicated and needs to be explored further in the future.

### 3.3.3 The influence of aerosols on HONO formation

As shown in Fig. S1, $HONO_{corr}/NO_2$ reached a pseudo-steady state from 03:00 to 6:00 LT every night. A correlation analysis of $HONO_{corr}/NO_2$ with $PM_{2.5}$ was carried out in the pseudo-steady state to understand the impact of aerosols on HONO production. Although we did not measure the aerosol surface density, the aerosol mass concentration can be used to replace this parameter (Huang et al., 2017;Park et al., 2004;Cui et al., 2018). The positive correlation of $HONO_{corr}$ with $PM_{2.5}$ ($R_1 = 0.54$) (Fig. 5a) may be a result of atmospheric physical processes such as convergence and diffusion. Using the $HONO_{corr}/NO_2$ ratio instead of a single HONO concentration for correlation analysis with $PM_{2.5}$ reduce the impact of physical processes and indicate the extent of conversion of $NO_2$ to HONO. Therefore, it was more credible that $HONO_{corr}/NO_2$ would be moderately positively correlated with $PM_{2.5}$ ($R_2 = 0.23$) during the whole observation period (Fig. 5b). As denoted by larger green squares in the figure, $HONO_{corr}/NO_2$ correlated well with $PM_{2.5}$ when its concentration was higher than 35 µg·m$^{-3}$ ($R_3 = 0.47$) (Fig. 5b). The larger the amount of HONO produced by the heterogeneous reaction of $NO_2$ on the aerosol surface, the better the correlation between $HONO/NO_2$ and $PM_{2.5}$ (Cui et al., 2018;Wang, 2003;Hou et al., 2016;Li et al., 2012;Nie et al., 2015).

### 3.4 Daytime sources of HONO

### 3.4.1 HONO photostationary-state approach

Having discussed the nighttime chemical behavior of HONO, we now concentrate on the daytime chemical behavior of HONO. A calculation of the photostationary state (PSS) was conducted to preliminarily assess HONO concentrations during the daytime, especially the influence of any potential additional sources. It is hoped that HONO is in the photostationary state in the daytime due to its production from oxidation of NO by OH (Reaction (R2)), reformation of OH and NO by rapid photolysis (Reaction (R3)), and oxidation of HONO itself by OH (Reaction (R4)).

$$NO + OH \rightarrow HONO \tag{R2}$$

$$HONO + h\nu \ (320\text{–}400nm) \rightarrow NO + OH \tag{R3}$$

$$HONO + OH \rightarrow NO_2 + H_2O \tag{R4}$$

The photostationary concentration $[HONO]_{PSS}$ was estimated by

$$[HONO]_{PSS} = \frac{k_{OH+NO}[OH][NO]}{k_{OH+HONO}[OH] + J(HONO)}, \tag{5}$$



in which $k_{OH+NO} = 7.4 \times 10^{-12}$ cm$^3$ molecules$^{-1}$ s$^{-1}$ and $k_{OH+HONO} = 6.0 \times 10^{-12}$ cm$^3$ molecules$^{-1}$ s$^{-1}$, values taken from a previous study (Sörgel et al., 2011). The OH concentration ([OH]) was estimated in this study because no data for this value were available. An improved empirical formula, Eq. (6), was applied to estimate [OH] using the NO$_2$ and HONO concentrations and the photolysis rate constants ($J$) of NO$_2$, O$_3$, and HONO (Wen et al., 2019).

$$[OH] = 4.1 \times 10^9 \times \frac{J(O^1D)^{0.83} \times J(NO_2)^{0.19} \times (140 \times NO_2 + 1) + HONO \times J(HONO)}{0.41 \times NO_2^2 + 1.7 \times NO_2 + 1 + NO \times k_{NO+OH} + HONO \times k_{HONO+OH}} \tag{6}$$

During spring, summer, autumn, and winter, the average midday OH concentrations were $3.89 \times 10^6$ cm$^{-3}$, $1.36 \times 10^7$ cm$^{-3}$, $1.07 \times 10^7$ cm$^{-3}$, and $2.97 \times 10^6$ cm$^{-3}$, respectively, which were within the range of those obtained in other studies varying from $0.5 \times 10^6$ cm$^{-3}$ to $4 \times 10^6$ cm$^{-3}$ in winter (Wang et al., 2017a) and from $1 \times 10^7$ cm$^{-3}$ to $2 \times 10^7$ cm$^{-3}$ in summer (Li et al., 2012).

Clearly, the [HONO]$_{PSS}$ values cannot reproduce the daytime HONO concentration (Fig. 6). The [HONO]$_{PSS}$ does, however, replicate the diurnal variations of HONO, peaking in the morning rush hour (08:00-10:00 LT), as characterized by the high NO concentration. However, [HONO]$_{PSS}$ values reduced to zero at 17:00–18:00 LT after this morning peak, which was caused by the photolysis rate of HONO being notably faster than the rate of the only source from Reaction (R2). The value of [HONO]$_{PSS}$ showed different levels in the four seasons, with the highest values in winter and the lowest values in summer due to seasonal variation of the photolysis rate constant. This indicates that the largest unknown sources appeared in summer while the smallest unknown sources appeared in winter. This result is consistent with the quantitative results regarding daytime unknown sources, as will be presented in the next section. The [HONO]$_{PSS}$ values correlated well with NO concentrations ($R = 0.865$), while the correlations with [OH] ($R = -0.081$) and $J$(HONO) ($R = -0.072$) were weak. Therefore, the HONO gas-phase chemistry was dominated by the availability of NO. Simply considering the homogeneous gas-phase reaction is far from matching the observed HONO concentrations. The HONO values calculated based on PSS were more than an order of magnitude smaller than the observed daytime HONO values, suggesting significant unknown sources for HONO, while the gas-phase reaction (Reaction (R3)) only accounts for a small proportion of the observed values, especially in summer.

### 3.4.2 Budget analysis of HONO

From the analysis in Sect. 3.4.1, it appears that there are additional sources of HONO in the daytime, because the [HONO]$_{PSS}$ value is much lower than the observed HONO concentration. Here, $R_{unknown}$ is used to stand for the additional sources. The value of $R_{unknown}$ was estimated based on the balance between sources and sinks due to its short atmospheric lifetime. The sources are: (1) oxidation of NO by OH ($R_{OH+NO} = k_{OH+NO}[NO][OH]$), (2) dark heterogeneous production ($P_{het}$), and (3) direct vehicle emission ($P_{emis}$); the sinks are (1) HONO photolysis ($R_{phot} = J_{HONO}[HONO]$), (2) oxidation of HONO by OH ($R_{OH+HONO} = k_{OH+HONO}[HONO][OH]$), and (3) dry deposition ($L_{dep}$). The value of $R_{unknown}$ can then be calculated according to





$$R_{\text{unknown}} = J_{\text{HONO}}[\text{HONO}] + k_{\text{OH+HONO}}[\text{HONO}][\text{OH}] + L_{\text{dep}} + \frac{\Delta[\text{HONO}]}{\Delta t} - k_{\text{OH+NO}}[\text{NO}][\text{OH}] - P_{\text{het}} - P_{\text{emis}}, \tag{7}$$

where $\frac{\Delta[\text{HONO}]}{\Delta t}$ is the observed change of HONO concentration (ppb·s$^{-1}$). The value of $\frac{\Delta[\text{HONO}]}{\Delta t}$ is the concentration difference between the center of one interval (1 min) and the center of the next interval, and this accounts for changes in concentration

levels (Sörgel et al., 2011). The parameter $L_{\text{dep}}$ can be quantified by multiplying the dry deposition rate of HONO by the observed HONO concentration and then dividing by the mixing layer height ( $L_{\text{dep}} = \frac{v_{\text{HONO}}^{\text{ground}} \times [\text{HONO}]}{H}$ ). A value of $v_{\text{HONO}}^{\text{ground}} = 2 \text{ cm·s}^{-1}$ was used for the deposition rate (Sörgel et al., 2011;Su et al., 2008b). The mixing layer heights during spring, summer, autumn, and winter were 1074.4 m, 1173.8 m, 1494.6 m, and 1310.4 m, respectively (Gao, 1999). In summarizing the known HONO sources, we included the nighttime heterogeneous production as a known source based on the

assumption that the day continues in the same way as the night (Sörgel et al., 2011). The term $P_{\text{het}}$ was parameterized by NO$_2$ conversion at night using the formula $P_{\text{het}} = C_{\text{HONO}}^{\text{C}}[\text{NO}_2]$ (Alicke, 2002).

Figure 7 shows the contributions of each term in Eq. (6) to the HONO budgets in different seasons. Photolysis of HONO ($R_{\text{phot}}$) formed the largest proportion of the sinks in all four seasons, accounting for 94.69%, 96.85%, 96.10%, and 95.01% in spring, summer, autumn, and winter, respectively. The value of $R_{\text{phot}}$ in summer was the highest (10.69 ppb·h$^{-1}$) and this was 4.95,

2.29, and 5.85 times higher than that in spring, autumn, and winter, respectively. The oxidation of HONO by OH contributed little to HONO sinks (2.49% of all sinks). Dry deposition ($L_{\text{dep}}$) was also very small (1.85% of all sinks). As for known sources, $R_{\text{OH+NO}}$ was the main known source in all four seasons, wherein the largest proportion was found in summer (80.73%), followed by autumn (70.98%), winter (66.27%), and spring (51.48%). Direct emission was second among the known sources, accounting for 40.83%, 15.78%, 23.55%, and 30.10% in spring, summer, autumn, and winter, respectively. Dark

heterogeneous formation ($P_{\text{hete}}$) was almost negligible in the daytime, accounting for approximately 5.07% of known sources during the whole observation period. As for unknown sources, these made up the largest proportion of all sources found in summer (84.92%), followed by autumn (80.29%) and spring (49.98%). However, the unknown sources only accounted for 15.26% of all sources in winter. This indicates that known sources of HONO can explain the majority of sources in winter, and it is not necessary to analyze the unknown sources in this season.

The values of $R_{\text{OH+NO}}$ in different seasons all reached their maximum in the morning, and this was followed by a gradual decrease. This parameter made up the highest proportion of all sources (56.15%) in winter, followed by spring (25.75%), autumn (13.99%), and summer (12.17%). In winter, with its low light intensity and high NO concentration, the homogeneous gas-phase reaction between NO and OH accounted for the majority of the daytime HONO sources. It is worth noting that $R_{\text{unknown}}$ exhibited a maximum at noon in all seasons except for winter. A previous study in Wangdu (Liu et al., 2019a) also

found that unknown sources of HONO reached a maximum at midday, with the strongest photolysis rates in summer. In the present study, the highest $R_{\text{unknown}}$ value at noon was 14.79 ppb·h$^{-1}$ in summer, followed by 6.49 ppb·h$^{-1}$ in autumn and 2.18 ppb·h$^{-1}$ in spring. The $R_{\text{unknown}}$ value peaked at 08:00 in winter, reaching 1.55 ppb·h$^{-1}$. This indicates that this source



depends on the season, strengthening the validity of the assumption that the missing HONO formation mechanism is related
to a photolytic source (Michoud et al., 2014). The magnitudes of these additional sources were much higher than those found
in Beijing (Wang et al., 2017a) (1.3–3.82 ppb·h⁻¹), in Guangzhou (0.77 ppb·h⁻¹) (Li et al., 2012), and in Xinken (~5 ppb·h⁻¹)
(Su et al., 2011).

### 3.4.3 Exploration of possible unknown daytime sources

According to the analyses in Sect. 1 and Sect. 3.4.2, the unknown sources are likely to be related to light. It was indeed found
that the unknown sources have a good correlation with the parameters related to light. It was reported in previous studies that
particulate nitrate photolysis is a source of HONO (Ye et al., 2017;Ye et al., 2016;Scharko et al., 2014;Romer et al.,
2018;McFall et al., 2018). We will discuss the possibility of HONO being produced by photolysis of particulate nitrate
($J(NO_3\_R) \times pNO_3^-$) at this site in the next section. There was a logarithmic relationship showing good correlation between
$R_{unknown}$ (ppb·h⁻¹) and $J(NO_3^-\_R) \times pNO_3^-$ (µg·m⁻³·s⁻¹) in spring ($R^2 = 0.6348$), summer ($R^2 = 0.7266$), and autumn
($R^2 = 0.5041$) (Fig. 8). In conditions of relatively lower $J(NO_3\_R) \times pNO_3^-$, $R_{unknown}$ increased rapidly with increasing $pNO_3^-$
concentration and its photolysis rate constant but reached a plateau after a critical value ($J(NO_3\_R) \times pNO_3^- > 1$ µg·m⁻³·s⁻¹ in
autumn, and $J(NO_3\_R) \times pNO_3^- > 2$ µg·m⁻³·s⁻¹ in spring and summer). This indicated that in conditions that were relatively
cleaner, the missing daytime source of HONO was limited by the $pNO_3^-$ concentration and the photolysis rate constant.
However, with severe haze or strong photolysis rate providing sufficient precursor or enough light to stimulate the reaction,
the HONO production did not increase as $J(NO_3\_R) \times pNO_3^-$ increased. It was found in a previous study (Scharko et al., 2014)
that $NO_2$ produced by $NO_3^-$ photolysis in situ is more easily absorbed by acidic solutions than the original gaseous $NO_2$.
Therefore, we found the limiting factor for $R_{unknown}$ to be the aerosol neutralization degree $F$ in spring, summer, and autumn.
Here, $F$ was calculated from the equivalent concentrations of ammonium, sulfate, and nitrate (Wang et al., 2015) such that

$$F = [NH_4^+]/(2 \times [SO_4^{2-}] + [NO_3^-]). \tag{8}$$

Considering the acidity of aerosols, we found that $R_{unknown}$ was limited when the aerosols were alkaline ($F > 1$). This field
observation validates laboratory research on the release of HONO from photolysis of $NO_3^-$ in acidic solutions (Scharko et al.,
2014).

We discuss whether photolysis of particulate nitrate is able to provide enough additional HONO by estimating the rate of
HONO production by nitrate photolysis (Zhou et al., 2007;Li et al., 2012;Wang et al., 2017a) using

$$J_{NO_3^- \to HONO} = \frac{R_{unknown} \times H}{f \times [NO_3^-] \times v_{NO_3^-} \times t_d}, \tag{9}$$

where $J_{NO_3^- \to HONO}$ is the rate of photolysis of $NO_3^-$ to form HONO, $v_{NO_3^-}$ is the dry deposition rate of $NO_3^-$ during the period
$t_d$, and $f$ is the proportion of the surface exposed to the sun at midday. Here, we suppose that the surfaces involving $NO_3^-$ were
exposed to light by a factor $f = 1/4$, taking mixing height $H = 250\,m$, $v_{NO_3^-} = 5$ cm·s⁻¹ over $t_d = 24$ h. We use the mean





value of $R_{\text{unknown}} = 2.36\ \mu g \cdot m^{-3} \cdot h^{-1}$ and $[NO_3^-] = 9.99\ \mu g \cdot m^{-3}$ in spring; $R_{\text{unknown}} = 15.25\ \mu g \cdot m^{-3} \cdot h^{-1}$ and $[NO_3^-] = 2.44\ \mu g \cdot m^{-3}$ in summer; and $R_{\text{unknown}} = 8.15\ \mu g \cdot m^{-3} \cdot h^{-1}$ and $[NO_3^-] = 3.73\ \mu g \cdot m^{-3}$ in autumn. The photolysis rates $J_{NO_3^- \rightarrow HONO}$ derived from

Eq. (8) were $1.52 \times 10^{-5}\ s^{-1}$, $4.02 \times 10^{-4}\ s^{-1}$, and $1.40 \times 10^{-4}\ s^{-1}$ for spring, summer, and autumn, respectively. These values are in the range $6.2 \times 10^{-6}$ to $5.0 \times 10^{-4}$ obtained in a previous study (Ye et al., 2017), which indicated that particulate nitrate photolysis was the main source in spring, summer, and autumn. The variability of $J_{NO_3^- \rightarrow HONO}$ may be caused by chemical composition, acidity, light-absorbing constituents, and the optical and other physical properties of aerosols.

### 3.5 Parameterization of HONO

Through an empirical parameterized formula, we can explore an accurate parameterization method for HONO, discuss the main control factors for the HONO concentration and its chemical behavior, and quantify its main sources and key kinetic parameters. As mentioned in Sect. 3.1, the HONO/NO$_x$ ratio is better than HONO/NO$_2$ as an indicator of HONO generation. In another study (Elshorbany et al., 2012), data were collected from 15 field observations all over the world to establish the correlation between the HONO/NO$_x$ ratio and the HONO concentration in global models. Therefore, we applied this method

in this study to parameterize the HONO concentration. As shown in Fig. 9, the HONO/NO$_x$ ratios in the four seasons were close to the calculated value (0.02). However, there were seasonal variations in the slope, showing a maximum in summer ($2.60 \times 10^{-2}$), followed by autumn ($2.06 \times 10^{-2}$), and a minimum in winter ($1.59 \times 10^{-2}$). Except for in spring, HONO showed good correlation with NO$_x$, with $R^2$ values ranging from 0.8972 to 0.9621. Therefore, we used slopes of $2.60 \times 10^{-2}$, $2.06 \times 10^{-2}$, and $1.59 \times 10^{-2}$ to parameterize the HONO concentrations in summer, autumn, and winter, respectively. As for spring, though

only a weak correlation between HONO and NO$_x$ was found, the majority of the HONO/NO$_x$ ratios fluctuated round a slope of 0.02 because concentrations of NO$_x$ greater than 60 ppb only accounted for 8.83% of the data. Therefore, a slope of 0.02 was applied in spring to parameterize the HONO concentration.

As can be seen from Fig. 10, the estimated values are very close to the observed values in the nighttime in autumn. After sunrise and before noon, the values observed were higher than the estimated values, and this difference gradually increases.

After noon and before sunset, the values observed were still higher than the values estimated, but the difference gradually decreases. This phenomenon was also found in the daytime in spring, summer and autumn, but not in winter. Compared with the daytime, the estimated values during the nighttime were closer to the observed values in both trend and value in all four seasons, which further demonstrates that nighttime HONO is mainly produced from the heterogeneous reaction of NO$_2$ on the ground and the surfaces of aerosols. Therefore, we should pay much more attention to simulation in the daytime. We

distinguish two main sectors, nighttime and daytime, to analyze the factors affecting the HONO diurnal variation (Liu, 2017). Although $J(HONO) \times HONO$ also correlated well with $J(NO_2) \times NO_2$ in all four seasons in this study and the linear fitting coefficients fluctuated around 0.01 in all four seasons (Fig. S2), bad simulation results during the daytime were found (Fig. S3) using

$$[HONO] = 0.01 \times [NO_2] \times J(NO_2)/J(HONO). \tag{10}$$





In contrast, excellent simulation results were found in a previous study using the same formula (Liu, 2017), which suggests that using the same simulation formula in different regions may obtain greatly varying results.

As discussed in Sect. 3.4.3, nitrate photolysis is the main source of HONO in spring, summer, and autumn during the daytime, while the homogeneous gas-phase reaction of NO and OH is the major source of daytime HONO in winter. Therefore, we take the photolysis of nitrate into the spring, summer, and autumn calculations, but we use the reaction of NO and OH in the

calculations for winter. In this way, the daytime simulation results are significantly improved (Fig. 10). This further demonstrates that the apportionment of HONO sources is credible.

### 3.6 Comparison of contributions of HONO and $O_3$ to OH radicals

Comparing the OH radical production via photolysis of HONO and $O_3$, the effect of the high HONO concentrations in the daytime on the tropospheric oxidation capacity was evaluated (Ryan et al., 2018). Nitrous acid is considered to be a crucial

source of OH radicals (Lee et al., 2016). As shown in Eq. (11), OH production rates from $O_3$ photolysis ($P_{OH}(O_3)$) were calculated based on $[O_3]$, $J(O^1D)$, and $[H_2O]$ (Liu et al., 2019c). Only $O(^1D)$ atoms produced by the $O_3$ photolysis at UV wavelengths less than 320 nm (Reaction (R5)) can combine with water to generate OH radicals (Reaction (R6)) in the atmosphere. The absolute water concentration was derived from temperature and RH. The reaction (R7) rates for $N_2$ is $3.1 \times 10^{-11}$ cm$^3$ molecules$^{-1}$ s$^{-1}$ and for $O_2$ is $4.0 \times 10^{-11}$ cm$^3$ molecules$^{-1}$ s$^{-1}$. The net OH formation from HONO was estimated

by Eq. (12) (Su et al., 2008b;Sörgel et al., 2011;Li et al., 2018;Atkinson et al., 2004).

$$P_{OH}(O_3) = 2J(O^1D)[O_3]\phi OH, \quad \phi OH = k_6[H_2O]/(k_6[H_2O] + k_7[M]) \tag{11}$$

$$O_3 + hv \rightarrow O(^1D) + O_2 \ (hv < 320 \text{ nm}) \tag{R5}$$

$$O(^1D) + H_2O \rightarrow 2OH \tag{R6}$$

$$O(^1D) + M \rightarrow O(^3P) + M \ (M \text{ is } N_2 \text{ or } O_2) \tag{R7}$$

$$P_{OH}(HONO) = J_{HONO}[HONO] - k_{OH+NO}[NO][OH] - k_{OH+HONO}[HONO][OH] \tag{12}$$

The diurnal patterns of $P(OH)$ are shown in Fig. 11. The formation rates of OH from $O_3$ photolysis peaked in midday at around 0.71 ppb·h$^{-1}$, 5.80 ppb·h$^{-1}$, 2.21 ppb·h$^{-1}$, and 0.48 ppb·h$^{-1}$ for spring, summer, autumn, and winter, respectively. The variation of $P_{OH}(O_3)$ is consistent with $J(O^1D)$ (Fig. S4), peaking in midday and in summer on a diurnal and a seasonal timescale, respectively. For summer and autumn, $P_{OH}(HONO)$ had a similar trend as $P_{OH}(O_3)$, peaking at around noon at the time of the

highest $J(HONO)$, but this was negligible at sunrise and sunset (Fig. S5). For spring and winter, however, $P_{OH}(HONO)$ reached a maximum in the morning rush hour caused by the combined influences of high HONO concentration and high $J(HONO)$. A similar result was also found in southwest Spain from mid-November to mid-December 2008 (Sörgel et al., 2011). These results show that HONO contributes considerably to the morning atmospheric oxidizing capacity of the suburban atmosphere of Xiamen. Although HONO concentrations (average: 0.66 ppb) are much lower than $O_3$ concentrations (average: 32.02 ppb)





during 07:00–16:00 LT, daytime HONO photolysis forms significantly more OH than daytime photolysis of $O_3$ in all four seasons. Generally, the mean value of $P_{OH}$(HONO) from 07:00 to 16:00 LT was 4.31 ppb·h$^{-1}$, and the average $P_{OH}(O_3)$ was 1.14 ppb·h$^{-1}$. This indicates that HONO concentrations at 0.66 ppb during 07:00–16:00 LT increase the formation of OH radicals by an order of magnitude, greatly increasing the local daytime tropospheric oxidative capacity. A similar result was found in Melbourne, where the peak OH production rate reached 2 ppb·h$^{-1}$ from 0.4 ppb HONO (Ryan et al., 2018). The

important role of HONO in the production of OH promotes photochemical peroxyacetyl nitrate formation (Hu et al., 2020).

## 4. Conclusions

We conducted measurements of HONO in the atmosphere at an IUE supersite in a coastal city of southeastern China in August, October, and December 2018 and March 2019, finding an average HONO concentration of 0.54 ± 0.47 ppb across the whole observation period. Concentrations of HONO in spring and summer were higher than in winter and autumn, which was

consistent with seasonal variations in RH. Both higher HONO concentrations in the daytime and the HONO/NO$_x$ ratio peaking around noon suggested that additional, unknown sources of HONO might be related to light. It was found that the contribution from vehicle exhaust emissions (1.64%) was higher than that found in most other studies due to the site being surrounded by several expressways with a large number of passing diesel vehicles. The average nocturnal conversion rate of $NO_2$ to HONO was 0.47% h$^{-1}$, which was within the range 0.29–2.40% h$^{-1}$ found by other studies. The HONO$_{corr}$/NO$_2$ ratio increased with

RH and the concentration of PM$_{2.5}$ during the nighttime, which indicates that nocturnal heterogeneous reactions on the surfaces of aerosols are the major source of HONO. However, dark heterogeneous formation ($P_{hete}$) was almost negligible in the daytime, accounting for approximately 5.07% of known sources across the whole observation period. Observed values in the daytime were up to 50 times higher than those calculated from the PSS, suggesting that there were a large number of daytime sources of HONO. The highest proportion of all sources was $R_{OH+NO}$ in winter (56.15%), while $R_{unknown}$ made up at the largest

proportion of all sources in summer (84.92%), autumn (80.29%), and spring (49.98%). It was found that there was a logarithmic relationship between $R_{unknown}$ and particulate nitrate photolysis, and the limiting factor was particulate acidity in spring, summer, and autumn. The variation of HONO at night can be accurately simulated based on the HONO/NO$_x$ ratio, while the main sources should be considered for daytime simulation. Local tropospheric oxidation capacity was significantly increased by HONO during 07:00–16:00, providing an OH radical source (4.31 ppb·h$^{-1}$) an order of magnitude greater than its concentration

(0.66 ppb).

**Data availability.**

Measurement data at the IUE station, including HONO data and relevant trace gases and aerosol data as well as meteorological data, are available upon request from the corresponding author before the IUE database is open to the public.




**Authorship Contribution Statement**

Baoye Hu and Jun Duan contributed equally to this work. Baoye Hu and Jun Duan collected the HONO data and contributed to the data analysis. Baoye Hu wrote the manuscript. Baoye Hu, Jun Duan performed the experiments. Jun Duan and Fang Wu built equipment of IBBCEEAS. Youwei Hong, Min Qin and Jinsheng Chen revised manuscript. Min Qin, Pinhua Xie and Jinsheng Chen designed the manuscript. Jinsheng Chen supported funding of observation and research. Lingling Xu, Mengren Li, Yahui Bian contributed to discussions of results.

**Competing interests**

The authors declare that they have no conflict of interest.

**Acknowledgments**

This study was funded by the National Key Research and Development Program (2017YFC0209400, 2016YFC02005, 2016YFC0112200), the National Natural Science Foundation of China (41575146, 41875154), the FJIRSM&IUE Joint Research Fund (RHZX-2019-006), the Center for Excellence in Regional Atmospheric Environment, CAS (E0L1B20201), State Key Laboratory of Environmental Chemistry and Ecotoxicology, Research Center for Eco-Environmental Sciences, CAS and Xiamen Atmospheric Environment Observation and Research Station of Fujian Province.

**Appendix A. Supplementary information**

Attached please find supplementary information associated with this article.



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





**Figure Captions**

**Figure 1: Maps showing the position of Xiamen in China (left) and the position of the IUE supersite in Xiamen (right).**

**Figure 2: Diurnal variations in (a) HONO, (b) $NO_x$, (c) $HONO/NO_x$, and (d) $J(NO_2)$. The gray shading indicates nighttime (18:00–06:00, including 18:00).**

**Figure 3: Scatter plots of $NO_2$ versus HONO color coded by $J(NO_2)$. The three dashed lines represent 10%, 5%, and 1% ratios of $HONO/NO_2$. Daytime was 06:00–18:00 LT, including 06:00.**

**Figure 4: Scatter plots of nighttime $HONO_{corr}/NO_2$ ratios versus RH. The average top-five $HONO_{corr}/NO_2$ in every 5% RH interval are shown as orange squares, and the error bars show ±1 SD.**

**Figure 5: The correlation between $PM_{2.5}$ and $HONO_{corr}$ (left) and the correlation between $PM_{2.5}$ and $HONO_{corr}/NO_2$ (right). The squares depict $PM_{2.5} \geq 35 \ \mu g \cdot m^{-3}$; all scattered points are from the time when the ratio of $HONO_{corr}/NO_2$ reached a pseudo-steady state each night (03:00–06:00 LT).**

**Figure 6: Average diurnal variations of HONO concentrations observed (solid markers/lines) and $HONO_{PSS}$ calculated by Eq. (1) (hollow markers and dashed lines).**

**Figure 7: Average diurnal variations of each source (>0) and sink (<0) of HONO in the four seasons.**

**Figure 8: Relationships between the photolysis of particulate nitrate and $R_{unknown}$, colored by $F$ in spring, summer, and autumn.**

**Figure 9: The ratio of $HONO/NO_x$ in the four seasons (correlation between the average of $NO_x$ per 10 ppb interval and the average value of HONO).**

**Figure 10: The diurnal variations in the measured values of HONO (black squares), the estimated values of HONO using the parameterized formula (red circles), and the estimated values of HONO using the parameterized formula combined with the main daytime sources (green triangles).**

**Figure 11: Comparison of OH formation by photolysis of HONO and $O_3$ in the four seasons.**



**Figure 1: Maps showing the position of Xiamen in China (left) and the position of the IUE supersite in Xiamen (right).**





**Figure 2: Diurnal variations in (a) HONO, (b) NO$_x$, (c) HONO/NO$_x$, and (d) $J$(NO$_2$). The gray shading indicates nighttime (18:00–06:00, including 18:00).**




**Figure 3: Scatter plots of NO₂ versus HONO color coded by _J_(NO₂). The three dashed lines represent 10%, 5%, and 1% ratios of HONO/NO₂. Daytime was 06:00–18:00 LT, including 06:00.**







**Figure 4: Scatter plots of nighttime HONO$_{corr}$/NO$_2$ ratios versus RH. The average top-five HONO$_{corr}$/NO$_2$ in every 5% RH interval are shown as orange squares, and the error bars show ±1 SD.**






**Figure 5: The correlation between $PM_{2.5}$ and $HONO_{corr}$ (left) and the correlation between $PM_{2.5}$ and $HONO_{corr}/NO_2$ (right). The squares depict $PM_{2.5} \geq 35$ μg·m$^{-3}$; all scattered points are from the time when the ratio of $HONO_{corr}/NO_2$ reached a pseudo-steady state each night (03:00–06:00 LT).**





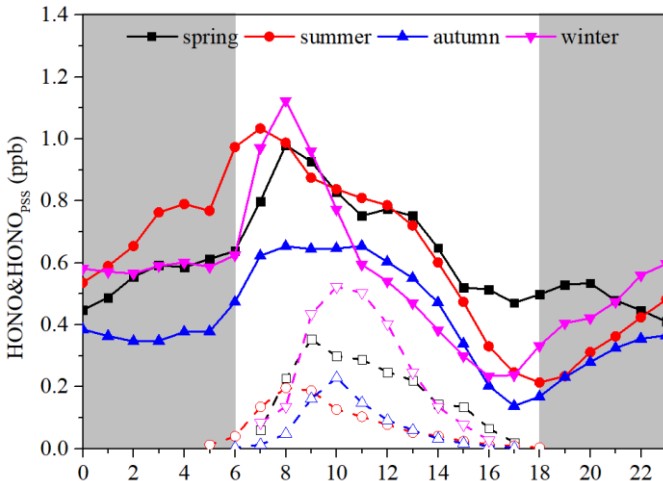

**Figure 6: Average diurnal variations of HONO concentrations observed (solid markers/lines) and HONO$_{PSS}$ calculated by Eq. (1) (hollow markers and dashed lines).**

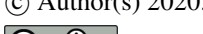





**Figure 7: Average diurnal variations of each source (>0) and sink (<0) of HONO in the four seasons.**





Figure 8: Relationships between the photolysis of particulate nitrate and $R_{unknown}$, colored by $F$ in spring, summer, and autumn.



**Figure 9: The ratio of HONO/NO$_x$ in the four seasons (correlation between the average of NO$_x$ per 10 ppb interval and the average value of HONO).**

**Figure 10:** **The diurnal variations in the measured values of HONO (black squares), the estimated values of HONO using the**
**parameterized formula (red circles), and the estimated values of HONO using the parameterized formula combined with the main**
**daytime sources (green triangles).**






**Figure 11: Comparison of OH formation by photolysis of HONO and O₃ in the four seasons.**





**Tables**

**Table 1.** Overview of the HONO and $NO_x$ concentrations measured in Xiamen and comparison with other measurements.

**Table 2.** Emission ratios of fresh vehicle plumes $\Delta HONO/\Delta NO_x$.

**Table 3.** Overview of the conversion frequencies from $NO_2$ to HONO in Xiamen and comparisons with other studies.




**Table 1.** Overview of the HONO and NO$_x$ concentrations measured in Xiamen and comparison with other measurements.

| Location | Date | HONO (ppb) | | NO$_2$ (ppb) | | NOx (ppb) | | HONO/NO$_2$ | | HONO/NOx | | Reference |
|---|---|---|---|---|---|---|---|---|---|---|---|---|
| | | Day | Night | Day | Night | Day | Night | Day | Night | Day | Night | |
| Xiamen/China(suburban) | Aug.2018-Mar.2019 | 0.63 | 0.46 | 13.6 | 16.3 | 20.9 | 19.9 | 0.061 | 0.028 | 0.046 | 0.024 | This work |
| | Mar.2019(spring) | 0.72 | 0.51 | 18.5 | 17.7 | 28.6 | 24.5 | 0.046 | 0.032 | 0.034 | 0.028 | |
| | Aug.2018(summer) | 0.72 | 0.51 | 11.0 | 15.7 | 16.6 | 18.9 | 0.094 | 0.031 | 0.072 | 0.027 | |
| | Oct.2018(autumn) | 0.50 | 0.33 | 11.4 | 14.3 | 14.1 | 15.1 | 0.060 | 0.023 | 0.048 | 0.022 | |
| | Dec.2018(winter) | 0.61 | 0.52 | 15.8 | 18.3 | 28.0 | 23.1 | 0.036 | 0.026 | 0.023 | 0.022 | |
| Jinan/China(urban) | Sep 2015-Aug 2016 | 0.99 | 1.28 | 25.8 | 31.0 | 40.6 | 46.4 | 0.056 | 0.079 | 0.035 | 0.040 | (Li et al., 2018) |
| | Sep.-Nov. 2015 (autumn) | 0.66 | 0.87 | 23.2 | 25.4 | 37.5 | 38.0 | 0.034 | 0.049 | 0.022 | 0.034 | |
| | Dec.2015-Feb.2016(winter) | 1.35 | 2.15 | 34.6 | 41.1 | 64.8 | 78.5 | 0.047 | 0.056 | 0.031 | 0.034 | |
| | Mar.-May 2016 (spring) | 1.04 | 1.24 | 25.8 | 35.8 | 36.0 | 47.3 | 0.052 | 0.046 | 0.041 | 0.035 | |
| | Jun.-Aug. 2016 (summer) | 1.01 | 1.20 | 19.0 | 22.5 | 25.8 | 29.1 | 0.079 | 0.106 | 0.049 | 0.060 | |
| Nanjing/China(suburban) | Nov. 2017-Nov. 2018 | 0.57 | 0.80 | 13.9 | 18.9 | 19.3 | 24.9 | 0.044 | 0.045 | 0.036 | 0.041 | (Liu et al., 2019c) |
| | Dec.-Feb. (winter) | 0.92 | 1.15 | 23.1 | 28.4 | 37.7 | 45.5 | 0.038 | 0.040 | 0.025 | 0.029 | |
| | Mar.-May (spring) | 0.59 | 0.76 | 12.9 | 17.4 | 15.9 | 19.1 | 0.049 | 0.048 | 0.042 | 0.046 | |
| | Jun.-Aug. (summer) | 0.34 | 0.56 | 7.7 | 12.5 | 9.1 | 13.5 | 0.051 | 0.048 | 0.045 | 0.046 | |
| | Sep.-Nov. (autumn) | 0.51 | 0.81 | 13.4 | 18.9 | 17.7 | 25.1 | 0.035 | 0.044 | 0.029 | 0.039 | |
| Hongkong/China | Aug.2011(summer) | 0.70 | 0.66 | 18.1 | 21.8 | 29.3 | 29.3 | 0.042 | 0.031 | 0.028 | 0.025 | (Xu et al., 2015) |



| | | | | | | | | | | | |
|---|---|---|---|---|---|---|---|---|---|---|---|
| | Nov.2011(autumn) | 0.89 | 0.95 | 29.0 | 27.2 | 40.6 | 37.2 | 0.030 | 0.034 | 0.021 | 0.028 | |
| | Feb.2012(winter) | 0.92 | 0.88 | 25.8 | 22.2 | 48.3 | 37.8 | 0.035 | 0.036 | 0.020 | 0.025 | |
| | May2012(spring) | 0.40 | 0.33 | 15.0 | 14.7 | 21.1 | 19.1 | 0.030 | 0.022 | 0.022 | 0.019 | |
| Guangzhou/China(urban) | Jun.2006 | 2.00 | 3.50 | 30.0 | 20.0 | - | - | 0.067 | 0.175 | - | - | (Qin et al., 2009) |
| Xi'an/China | Jul.-Aug.2015 | 1.57 | 0.51 | 24.7 | 15.4 | - | - | 0.062 | 0.033 | - | - | (Huang et al., 2017) |
| Santiago/Chile(urban) | Mar.-Jun.2005 | 1.50 | 3.00 | 20.0 | 30.0 | 40.0 | 200.0 | 0.075 | 0.100 | 0.038 | 0.015 | (Elshorbany et al., 2009) |
| Rome/Italy(urban) | May-Jun.2001 | 0.15 | 1.00 | 4.0 | 27.2 | 4.2 | 51.2 | 0.038 | 0.037 | 0.024 | 0.020 | (Acker et al., 2006) |
| Kathmandu/Nepal(urban) | Jan.-Feb.2003 | 0.35 | 1.74 | 8.6 | 17.9 | 13.0 | 20.1 | 0.041 | 0.097 | 0.027 | 0.087 | (Yu et al., 2009) |

Note: Night (18:00-6:00, including 18:00, local time); Day (6:00-18:00, including 6:00, local time)

$NOx=NO_2$ (IBBCEAS)+NO (Thermal 42i). IBBCEAS measure both HONO and $NO_2$. The $NO_2$ concentration is always overestimated by the Thermo Fisher 42i.





**Table 2.** Emission ratios of fresh vehicle plumes $\Delta HONO/\Delta NO_x$.

| Date | Time | $\Delta NO/\Delta NOx$ | $R^2$ | $\Delta HONO/\Delta NOx$ (%) |
|---|---|---|---|---|
| 2018/8/1 | 7:00-8:55 | 1.1621 | 0.6897 | 2.17 |
| 2018/8/8 | 5:40-5:55 | 0.8727 | 0.8023 | 2.69 |
| 2018/8/21 | 5:00-5:55 | 0.8571 | 0.7553 | 1.14 |
| 2018/8/22 | 7:20-7:45 | 0.4998 | 0.6151 | 4.76 |
| 2018/8/23 | 5:20-5:55 | 0.7321 | 0.8089 | 2.12 |
| 2018/8/23 | 6:00-6:55 | 0.8321 | 0.6687 | 2.19 |
| 2018/8/31 | 23:35-23:55 | 1.1861 | 0.8130 | 1.18 |
| 2018/10/23 | 1:05-1:25 | 0.9893 | 0.6566 | 1.27 |
| 2018/12/4 | 7:20-7:40 | 0.9594 | 0.8502 | 1.11 |
| 2018/12/10 | 11:00-11:15 | 0.8778 | 0.6735 | 1.79 |
| 2018/12/11 | 0:00-0:50 | 0.9424 | 0.6972 | 0.58 |
| 2018/12/11 | 1:25-1:55 | 0.8492 | 0.8237 | 1.26 |
| 2018/12/11 | 2:50-3:55 | 0.7405 | 0.7520 | 2.87 |
| 2018/12/11 | 4:00-4:55 | 0.9652 | 0.7686 | 2.12 |
| 2018/12/11 | 5:45-6:35 | 1.0243 | 0.6566 | 0.84 |
| 2018/12/11 | 6:40-7:40 | 0.9992 | 0.7067 | 1.59 |
| 2018/12/11 | 8:15-8:55 | 0.8333 | 0.6820 | 1.89 |
| 2018/12/13 | 7:00-8:50 | 0.8263 | 0.8127 | 1.02 |
| 2018/12/13 | 9:10-9:45 | 0.7235 | 0.7776 | 1.01 |
| 2018/12/16 | 7:00-7:55 | 0.7523 | 0.8939 | 0.98 |
| 2018/12/18 | 7:35-8:10 | 0.7046 | 0.7110 | 1.15 |
| 2018/12/20 | 22:50-23:10 | 0.9811 | 0.7736 | 0.97 |
| 2018/12/21 | 0:45-1:15 | 1.0029 | 0.8914 | 1.54 |
| 2018/12/22 | 6:40-7:35 | 1.0194 | 0.7010 | 2.36 |
| 2018/12/22 | 7:40-8:05 | 0.9932 | 0.7831 | 2.94 |
| 2018/12/25 | 21:00-22:10 | 0.9573 | 0.8857 | 1.64 |
| 2018/12/26 | 3:50-4:15 | 1.167 | 0.6540 | 1.39 |
| 2018/12/26 | 6:45-7:45 | 0.9971 | 0.8463 | 0.92 |
| 2018/12/26 | 7:55-8:25 | 0.9714 | 0.6919 | 2.95 |
| 2018/12/27 | 4:50-5:30 | 0.9365 | 0.7265 | 0.76 |
| 2019/3/6 | 7:30-8:05 | 1.0309 | 0.8283 | 0.74 |
| 2019/3/9 | 7:50-8:05 | 0.9933 | 0.9203 | 0.24 |
| 2019/3/9 | 12:00-12:55 | 0.9627 | 0.6444 | 0.51 |
| 2019/3/18 | 6:35-8:35 | 1.0382 | 0.6967 | 3.14 |





**Table 3.** Overview of the conversion frequencies from $NO_2$ to HONO in Xiamen and comparisons with other studies.

| Location | Date | Conversion rate (% h$^{-1}$) | Reference |
|---|---|---|---|
| Xiamen/China | Aug.2018-Mar.2019 | 0.47 | This study |
| | Mar.2019(spring) | 0.47 | |
| | Aug.2018(summer) | 0.55 | |
| | Oct.2018(autumn) | 0.48 | |
| | Dec.2018(winter) | 0.37 | |
| Xinken/China | Oct.-Nov.,2004 | 1.60 | (Su et al., 2008a) |
| Jinan/China | Sep.,2015-Aug.,2016 | 0.68 | (Li et al., 2018) |
| | Mar.-May 2016(spring) | 0.43 | |
| | Jun.-Aug. 2016(summer) | 0.69 | |
| | Sep.-Nov. 2015(autumn) | 0.75 | |
| | Dec.2015-Feb. 2016(winter) | 0.83 | |
| Guangzhou/China | Jun.,2006 | 2.40 | (Li et al., 2012) |
| Spain | Nov.-Dec.,2008 | 1.50 | (Sörgel et al., 2011) |
| Beijing/China | Sep.2015-July 2016 | 0.80 | (Wang et al., 2017a) |
| | Apr.-May, 2016 (spring) | 0.50 | |
| | Jun.-Jul., 2016 (summer) | 1.00 | |
| | Sep.-Oct. 2015 (autumn) | 0.90 | |
| | Jan.2016 (winter) | 0.60 | |
| Shandong/China | Nov.2013-Jan.2014 | 0.29 | (Wang et al., 2015) |
| Shanghai/China | Aug.2010-Jun.2012 | 0.70 | (Wang et al., 2013) |
| Eastern Bohai Sea/China | Oct.-Nov., 2016 | 1.80 | (Wen et al., 2019) |
| Hongkong/China | Aug.2011-May, 2012 | 0.52 | (Xu et al., 2015) |
| Kathmandu/South Asia | Jan.-Feb.,2003 | 1.4 | (Yu et al., 2009) |