# Peer review of "Exploration of the atmospheric chemistry of nitrous acid in a coastal city of southeastern China: Results from measurements across four seasons"

_Atmospheric Chemistry and Physics, 2020_

## Referee Comment (RC1) · Anonymous Referee #1 · 1 Oct 2020

The manuscript "Exploration of the atmospheric chemistry of nitrous acid in a coastal city of southeastern China: Results from measurements across four seasons" by Baoye Hu et al. reports year-long observations of HONO together with gaseous, particulate, and meteorological parameters which are relevant for investigating HONO sources. The manuscript adds valuable informations on HONO concentration level and its temporal variation under costal condition. Discussions on the HONO sources and on the HONO impacts on OH radical production is however inline with the current understanding. I would recommend the publication if my following comments are well addressed.

[Figure]

**General comments**

There are plenty of published papers describing HONO measurements, most of which are also using the similar methodology to investigate HONO sources and draw similar conclusion that daytime HONO is mainly originated from photolysis of nitrates. In order to make the manuscript more valuable to the community, I would suggest the authors put more efforts on summarizing the findings on HONO production in different environments (e.g., inland or costal, downtown or suburban or rural, seasons, RH levels, $NO_3^-$ levels, etc.) and compare those with this work. I think the comprehensive data set shown in the manuscript would well support the comparison. The current comparisons listed in Table 1 and Table 3 are too general and not quite informative compared to those already shown in many other publications.

**Specific comments**

**Line 93 – 96, Page 4:** The author should make a clear description that the time needed for the sampling period and the later IC analysis on the MARGA system. When synchronize high time resolution data (i.e., HONO, NOx, J values) to MARGA data, it should be done exactly for the MARGA sampling period. This should also be clearly described.

**Line 110, Page 4:** I would recommend the Section 3.1 focus on reporting the measurement results, discussions on HONO sources and OH production by HONO photolysis can be moved to the following specific sections.

**Line 111 – 115, Page 4:** As mentioned in the General Comments, I suggest to make the comparison in terms of environmental conditions.

**Line 128 – 131, Page 5:** How about the sea salt concentration observed in this study? The argument is based on the assumption that HONO is mainly formed by photolysis of sea salt, which could not be confirmed at this moment.

**Line 134 – 135, Page 5:** It would be helpful to confirm this by using the measured J, NOx, RH, etc.

**Line 162 – 163, Page 6:** I could not follow the argument that higher HONO/NOx ratio indicate unknown daytime HONO sources. The authors should first describe what is "unknown" and if the observation could not been explained by the well accepted theory.

**Line 184, Page 6:** How is the duration of air masses been determined?

**Line 242, Page 8 – Line 243 Page 9:** Seeing from the summer plot in Fig. 4, most blue points lie in values below 0.03 when RH is above 90%. It looks to me that the trend of orange line is biased by only few data points which have high HONO to $NO_2$ ratio.

**Line 312, Page 11:** Should be "Eq. (7)" instead of "Eq. (6)".

**Section 3.5, Page 13 – 14:** First of all, there are various assumptions on HONO production pathways been made in the previous sections. I would be better to provide a full picture on how large of the each contribution to the HONO formation. Secondly, under the title "Parameterization", the reader could not even find a formula used for predicting HONO production or concentration. Moreover, why Eq. (10) is suitable in other place than in this work? Would the parameterization described in this work more

reasonable and can be better used in the future?

**Section 3.6, Page 14 – 15:** I think the authors should make a clear statement that they are evaluation the primary production of OH radical. As shown in many publications investigating HOx budgets, the production of OH during daytime is mainly by $HO_2 / RO_2 + NO$ reaction.

---

## Referee Comment (RC2) · Anonymous Referee #2 · 1 Oct 2020

The manuscript "Exploration of the atmospheric chemistry of nitrous acid in a coastal city of southeastern China: Results from measurements across four seasons" by Hu et al. provides observations and analysis of compounds important for improving understanding of tropospheric chemistry. The topic is important to many readers, and this study is closely related to a large number of papers that try to understand atmospheric HONO abundance and its impact on oxidants. The writing is clear, and the observations are sufficiently unique and comprehensive to provide new insights.

Regrettably, the utility of the manuscript is compromised by the omission of many im-

portant experimental details, such that the context and relevance of the measurements reported here cannot be ascertained. Additionally, there are several analyses that are incomplete or difficult to understand. Consequently, I cannot recommend this paper for publication. I do hope that these measurements will receive further examination and that a paper will be written that considers some of the suggestions below.

The methodology section is far too brief, and many critical details are absent. The reference (Duan et al, 2018) that describes the HONO instrument notes the importance of characterizing HONO transmission and production in inlets. There is no mention of any of the sampling inlets. NO2 readily converts to HONO on surfaces, but there is no way to assess the importance of this artifact without a thorough description of inlet length, material, flow, etc. Are filters used on the IBBCEAS to remove ambient aerosol, as in Min et al., 2016? If so, how often are they changed? What are the uncertainties for the aerosol and NO measurements? It would be helpful to show how the IBBCEAS and TEI NO2 measurements compare. Line 105 says the TEI "might actually include other active nitrogen compounds". Did it? This assertion should be tested, or at least referenced.

The measurement site isn't described adequately. The paper notes that a coastal location, land/sea breeze effects, vehicle exhaust emission, and contributions from diesel traffic are important for understanding HONO abundance, but none of these contributions are detailed here. The conclusion states that site was surrounded by expressways, but these are not detailed in the body of the text. How close are the expressways? Are there diurnal traffic patterns? Figure 1 gives a map, but it does not have sufficient detail to understand the sampling location. The figure should show latitude on an axis, clearly identify land and water, show major roads. And the map should use km rather than miles. The meteorology must be described. Is there a land/sea breeze effect here? What is the mixed layer height? Is the top of the building always within the mixed layer? I expect some nighttime measurements are capturing a residual layer of pollution that may have been processed for longer periods. How large

is the city, and what is the proximity of soils and tall buildings (proposed sources of HONO)? The motivation for this paper is that coastal cities have been undersampled, but none of the characteristics important to this coastal location are described.

Critical ancillary measurements are not adequately reported. What are the Ozone levels at night? The paper reports the average ozone for the entire study, but this doesn't reveal whether ozone is titrated at night, whether there is large ozone production during the day, and the photochemical environment of the measurement location. What is the temperature at this location? Figures 2 and 6 show that the length of day is the same for all seasons, but this can't be right. On line 99, please describe what you mean with _R and _M in the photolysis rate constants.

Several of the interpretations are difficult for me to understand and require further analysis. For example, line 264 says "It is hoped that HONO is in the photostationary state….", and from there, all calculations assume that is the case. The PSS assumption needs to be carefully examined. An analysis of measurements from a similar height on top of a building in Houston show that the PSS assumption may not be correct (Lee et al, Urban measurements of atmospheric nitrous acid: A caveat on the interpretation of the HONO photostationary state, JGR 2013). The Lee et al paper shows that the PSS assumption needs to be carefully examined to quantify the strength of an unknown HONO source. And this is especially true for measurements that are adjacent to major expressways. Table 2 shows fresh vehicle plumes measured during midday with HONO/NOx comparable to nighttime measurements, which may suggest that these plumes are not in PSS. What is HONO/NOx (PSS) for the daytime plumes in Table 2?

The HONO production rate from unknown sources reported here is gigantic: 14.78 ppb/h in summer, when it accounted for nearly all HONO production. This number should be compared with previous reports. Have such high values every been reported before? Ryan et al (referenced here) report 1 ppb/hr, and some studies have shown that summer daytime HONO and HONO/NOx can be explained without invoking any unknown source (Lee et al, Urban measurements of atmospheric nitrous acid: A

caveat on the interpretation of the HONO photostationary state, JGR 2013; Neuman et al., HONO emission and production determined from airborne measurements over the Southeast U.S., JGR 2016).

Several of the figures are difficult to understand. What are the red lines and dashed lines in figure 8? The logarithmic fits should be described, as they don't appear to encompass the data. It appears that the data could be just as easily fit with a line. What is the color scale on the right? What are the green squares in Figure 10? It would be helpful to keep a consistent color scale for the seasons. All of the figure captions should be expanded to explicitly identify every symbol and line shown on each figure. Labels and units must be included for every axis and colorscale (these are missing on figs 1, 2, 3, 6, 8).

I have trouble making sense of the concluding lines of the abstract and conclusion. The conclusion ends (lines 448-450) by stating that HONO provides an OH radical source (4.31 ppb/h) an order of magnitude greater than its concentration (0.66 ppb). I don't understand the comparison of a production rate with a concentration. The order of magnitude increase is also mentioned in the previous section, but I cannot see where this value comes from. The last line of the abstract states the study "draws a full picture of the sources of HONO. . ." But the vast majority of sources are unidentified. A more accurate statement might be that the HONO observations here do not identify the processes that determine HONO chemistry.
* * *

---

## Author Comment (AC1) · 3 Dec 2020

*AUTHOR RESPONSES IN BLUE ITALIC TEXT*

The manuscript "Exploration of the atmospheric chemistry of nitrous acid in a coastal city of southeastern China: Results from measurements across four seasons" by Baoye Hu et al. reports year-long observations of HONO together with gaseous, particulate, and meteorological parameters which are relevant for investigating HONO sources. The manuscript adds valuable information on HONO concentration level and its temporal variation under costal condition. Discussions on the HONO sources and on the HONO impacts on OH radical production is however in line with the current understanding. I would recommend the publication if my following comments are well addressed.

*Response: Thanks for your valuable comments and positive feedback. We have corrected this manuscript according to your suggestion. Below are the point-to-point responses to general and specific comments.*

**General comments**

There are plenty of published papers describing HONO measurements, most of which are also using the similar methodology to investigate HONO sources and draw similar conclusion that daytime HONO is mainly originated from photolysis of nitrates. In order to make the manuscript more valuable to the community, I would suggest the authors put more efforts on summarizing the findings on HONO production in different environments (e.g., inland or costal, downtown or suburban or rural, seasons, RH levels, NO3 levels, etc.) and compare those with this work. I think the comprehensive data set shown in the manuscript would well support the comparison. The current comparisons listed in Table 1 and Table 3 are too general and not quite informative compared to those already shown in many other publications.

*Response: We would like to thank the reviewer's valuable comments and effort in reviewing the manuscript. We have summarized the findings on HONO production in different environments and compared those with this work. Since many studies did not measure $NO_3$ concentration, we used the most frequently measured NO and $NO_2$ concentration instead. As shown in Table 1, the HONO concentration measured at this site was comparable to those measured at other suburban sites (Liu et al., 2019;Xu et al., 2015;Nie et al., 2015;Park et al., 2004), was obvious lower than those measured at urban sites and industrial site (Li et al., 2018;Yu et al., 2009;Hou et al., 2016;Qin et al., 2009;Wang et al., 2013;Shi et al., 2020;Spataro et al., 2013;Huang et al., 2017;Wang et al., 2017), and was obvious higher than those measured at marine background(Wen et al., 2019), Marine boundary layer(Ye et al., 2016), and coastal remote(Meusel et al., 2016). Besides, this table was put in "Supplementary Material" due to it`s length.*

Table 1. Comparison of HONO concentrations and related parameters at this site with other regions.

| Site | Country | Type | Seasons | RH (%) | T (℃) | NO/NO$_2$ | HONO (ppb) | Reference |
|------|---------|------|---------|--------|-------|-----------|------------|-----------|
| Jinan | China | Urban | Annual | 51.42 | 16.07 | 15.38/27.92 | 1.15 ± 1.07 | (Li et al., 2018) |
| | | | Spring | 56.67 | 16.77 | 11.33/29.67 | 1.16 ± 0.90 | |
| | | | Summer | 38.67 | 26.67 | 5.67/17.33 | 1.12 ± 0.93 | |
| | | | Autumn | 53.00 | 16.33 | 13.00/23.67 | 0.78 ± 0.60 | |
| | | | Winter | 59.67 | 3.00 | 31.17/36.33 | 1.71 ± 1.62 | |
| Kathmandu | Nepal | Urban | Winter | — | — | 3.16/14.14 | 1.55 | (Yu et al., 2009) |
| Beijing | China | Urban | Severe haze | — | — | 29.35/48.1 | 1.95 | (Hou et al., 2016) |
| | | | Clean | — | — | 5.2/18.85 | 0.72 | |
| Guangzhou | China | Urban | Summer | 77 | 31.2 | —/30.3 | ~2.8 | (Qin et al., 2009) |
| Shanghai | China | Urban | Annual | — | — | —/18.78 | 0.92 ± 0.57 | (Wang et al., 2013) |
| Changzhou | China | Urban | Spring | 53.7 ± 19.8 | 18.7 ± 4.8 | 8.2/22.9 | 1.55 ± 1.21 | (Shi et al., 2020) |
| Beijing | China | Urban | Summer | 56.79 | 28.27 | 6.44/31.70 | 1.45 ± 0.58 | (Spataro et al., 2013) |
| | | | Winter | 26.02 | 3.51 | 25.90/38.76 | 1.04 ± 0.73 | |
| Xi'an | China | Urban | Summer | — | — | —/20.9 | 1.12 ± 0.97 | (Huang et al., 2017) |
| Beijing | China | Urban | Annual | 43.34 | 16.01 | —/25.60 | 1.44 ± 1.33 | (Wang et al., 2017) |
| | | | Spring | 34.73 | 18.44 | —/25.97 | 1.05 ± 0.95 | |
| | | | Summer | 55.30 | 28.11 | —/19.21 | 1.38 ± 0.90 | |
| | | | autumn | 51.11 | 17.33 | —/32.91 | 2.27 ± 1.82 | |
| | | | winter | 30.48 | -3.57 | —/19.96 | 1.05 ± 0.89 | |
| Nanjing | China | Industrial | Winter | 68 | 6.1 | 7.97/23.9 | 1.32 ± 0.92 | (Zheng et al., 2020) |
| Touji Island | China | Marine background | Autumn | 74 | 14.2 | 0.5/5.3 | 0.20 ± 0.20 | (Wen et al., 2019) |
| The North Atlantic Ocean | — | Marine boundary layer | 5 July 2013 | — | — | — | 0.0113 ± 0.0016 | (Ye et al., 2016) |
| | | | 8 July 2013 | — | — | — | 0.0088 ± 0.0023 | |

| | | | | | | | | |
|---|---|---|---|---|---|---|---|---|
| Cyprus | Mediterranean Sea | Coastal remote | summer | — | 18-28 | 0.02/0.14 | 0.035 ± 0.025 | (Meusel et al., 2016) |
| Kwangju | Korea | Suburban | Autumn | 74.55 | 15.08 | — | 0.67 ± 0.60 | (Park et al., 2004) |
| Nanjing | China | Suburban | Annual | 72 | 17.00 | 5.7/16.4 | 0.69 ± 0.58 | (Liu et al., 2019) |
| | | | Spring | 73 | 17.67 | 2.35/15.15 | 0.68 ± 0.48 | |
| | | | Summer | 77 | 28.00 | 1.2/10.1 | 0.45 ± 0.37 | |
| | | | Autumn | 72 | 18.11 | 5.25/16.15 | 0.66 ± 0.53 | |
| | | | Winter | 66 | 4.33 | 15.58/25.75 | 1.04 ± 0.75 | |
| Hongkong | China | Suburban | Annual | 73 | 25 | 10.7/21.7 | 0.71 | (Xu et al., 2015) |
| | | | Spring | 75 | 28 | 5.5/15.5 | 0.35 | |
| | | | Summer | 71 | 32 | 8/19.8 | 0.65 | |
| | | | Autumn | 67 | 23 | 10.1/26.8 | 0.93 | |
| | | | Winter | 78 | 17 | 19.3/24.7 | 0.91 | |
| Western Yangtze River delta | China | Suburban | Spring | — | — | — | 0.76 ± 0.79 | (Nie et al., 2015) |
| Xiamen | China | Suburban | Annual | 78.35 | 22.95 | 5.80/14.99 | 0.54 ± 0.47 | This work |
| | | | Spring | 84.21 | 16.59 | 8.47/18.10 | 0.62 ± 0.58 | |
| | | | Summer | 84.12 | 30.00 | 4.79/13.39 | 0.61 ± 0.39 | |
| | | | Autumn | 69.55 | 24.02 | 2.18/12.88 | 0.41 ± 0.30 | |
| | | | Winter | 78.13 | 18.41 | 8.86/17.03 | 0.54 ± 0.47 | |

Note: "—" means no data found in the corresponding reference.

**Specific comments**

**Line 93 – 96, Page 4:** The author should make a clear description that the time needed for the sampling period and the later IC analysis on the MARGA system. When synchronize high time resolution data (i.e., HONO, NOx, J values) to MARGA data, it should be done exactly for the MARGA sampling period. This should also be clearly described.

*Response: Thanks for your valuable suggestion. We have made a clear description that the time needed for the sampling period and the later IC analysis on the MARGA system. The inorganic composition of $PM_{2.5}$ aerosols ($SO_4^{2-}$, $NO_3^-$, $Cl^-$, $Na^+$, $NH_4^+$, $K^+$, $Ca^{2+}$, $Mg^{2+}$) and concentrations of gases (HONO, $HNO_3$, HCl, $SO_2$, $NH_3$) were determined using a Monitor for AeRosols and Gases in ambient Air (MARGA, Model ADI 2080, Applikon Analytical B.V., the Netherlands) consists of two identical sample boxes and one analytical box. Ambient air was drawn at the flow rate of 1 $m^3 \cdot h^{-1}$ into the sample box by a $PM_{2.5}$ cyclone (Teflon coated inlet, URG-2000-30ENB). Sample air was first drawn through the Wet Rotating Denuder (WRD) where water-soluble gases diffused to the absorption solution, then particles were collected in a Steam Jet Aerosol Collector (SJAC). Absorption solutions were drawn from the WRD and the SJAC to syringes (25 ml) in the analytical box. Each hour after the syringes had been filled, samples were injected to Metrohm anion (250 µl loop) and cation (500 µl loop) chromatographs with the internal standard (LiBr) for 15 minutes(Makkonen et al., 2012). Specific descriptions of the SJAC can be found in previous reports (Slanina et al., 2001;Wyers et al., 1993). Therefore, the times needed for the sampling period and the latter IC analysis on the MARGA system are a full hour and 15 minutes, respectively. The value measured in this hour is actually the concentration sampled in the previous hour, so the time corresponding to the sampling is matched with other instrument parameters (i.e., HONO, NOx, J values).*

**Line 110, Page 4:** I would recommend the Section 3.1 focus on reporting the measurement results, discussions on HONO sources and OH production by HONO photolysis can be moved to the following specific sections.

*Response: Thanks for your careful working. This way of writing is more logical. Some conclusions based on observations are more convincing and avoid repeating the results. The more specific discussion on HONO source focused on the Sec 3.2, 3.3, 3.4, and OH production by HONO photolysis focused on the Sec 3.6.*

**Line 111 – 115, Page 4:** As mentioned in the General Comments, I suggest to make the comparison in terms of environmental conditions.

*Response: Thanks for your valuable suggestion. The comparison in terms of environmental conditions has been made as suggested in revised manuscript.*

**Line 128 – 131, Page 5:** How about the sea salt concentration observed in this study? The argument is based on the assumption that HONO is mainly formed by photolysis of sea salt, which could not be confirmed at this moment.

*Response: Thanks for your valuable suggestion. The sea salt concentration was 2.91 $\mu g \cdot m^{-3}$ during the day and 2.73 $\mu g \cdot m^{-3}$ during the night. And the sea salt concentration during the day is significantly higher than that during the night. This assumption was based on the reference that nitrate photolysis of sea salt particulate acts as a significant source of HONO(Kasibhatla et al., 2018).*

**Line 134 – 135, Page 5:** It would be helpful to confirm this by using the measured J, NOx, RH, etc.

*Response: This conclusion had been confirmed by J and NOx in the next paragraph and by RH in Sec. 3.3.2.*

**Line 162 – 163, Page 6:** I could not follow the argument that higher HONO/NOx ratio indicate unknown daytime HONO sources. The authors should first describe what is "unknown" and if the observation could not be explained by the well accepted theory.

*Response: Thanks for your valuable suggestion. This conclusion was based on the reference that the large HONO/NOx observed at around noon revealed additional daytime source(s) of HONO(Xu et al., 2015;Liu et al., 2019). If the HONO sources during the daytime are consistent with those at night, the minimum HONO/NOx ratio should occur at noon due to the intense photochemical loss of HONO. Therefore, there must be additional sources of HONO during daytime. "Unknown" has been changed into "additional" in order to avoid misunderstanding.*

**Line 184, Page 6:** How is the duration of air masses been determined?

*Response: Thanks for your careful working. The meaning of the text is that the air mass meets the conditions (1), (3), (4) and (5), and the duration of the air mass cannot exceed 2 h (Liu et al., 2019;Xu et al., 2015), which was based on the following two reasons. Firstly, fresh air mass should be short time. An Air mass with high NOx lasting long could not be a local fresh air mass, but an aged air mass transporting from high NOx region, such as city region. Secondly, if the duration of air mass was too long, the HONO observed was easily affected by secondary production, which would overestimate vehicle emission.*

**Line 242, Page 8 – Line 243 Page 9:** Seeing from the summer plot in Fig. 4, most blue points lie in values below 0.03 when RH is above 90%. It looks to me that the trend of orange line is biased by only few data points which have high HONO to $NO_2$ ratio.

*Response: Thanks for your careful working. It is true that most blue points lie in values below 0.03 when RH is above 90%. However, top-five $HONO_{corr}/NO_2$ ratios would reduce the influence of those circumstances such as advection, the time of the night, and the surface density(Li et al., 2012;Stutz et al., 2004).Therefore, top-five $HONO_{corr}/NO_2$ ratios were applied to replace all ratios.*

**Line 312, Page 11:** Should be "Eq. (7)" instead of "Eq. (6)".

*Response: Revised.*

**Section 3.5, Page 13 – 14:** First of all, there are various assumptions on HONO production pathways been made in the previous sections. I would be better to provide a full picture on how large of the each contribution to the HONO formation. Secondly, under the title "Parameterization", the reader could not even find a formula used for predicting HONO production or concentration. Moreover, why Eq. (10) is suitable in other place than in this work? Would the parameterization described in this work more reasonable and can be better used in the future?

*Response: Thanks for your valuable suggestion. Firstly, the contribution of each HONO production pathways has been specifically displayed in Section 3.4.2. Secondly, although the formula used for predicting HONO concentration in this study had not been explicitly given, we had described in the manuscript what kind of method can be used to improve parameterize., such as "Therefore, we used slopes of $2.60 \times 10^{-2}$, $2.06 \times 10^{-2}$, and $1.59 \times 10^{-2}$ to parameterize the HONO concentrations in summer, autumn, and winter, respectively. As for spring, though only a weak correlation between HONO and NOx was found, the majority of the HONO/NOx ratios fluctuated round a slope of 0.02 because concentrations of NOx greater than 60 ppb only accounted for 8.83% of the data. Therefore, a slope of 0.02 was applied in spring to parameterize the HONO concentration.", and*

*"Therefore, we take the photolysis of nitrate into the spring, summer, and autumn calculations, but we use the reaction of NO and OH in the calculations for winter. In this way, the daytime simulation results are significantly improved". Eq. (10) ([HONO] = 0.01×[NO$_2$] × J(NO$_2$)/J(HONO)) can be regarded as a combination of [NO$_2$] with J(NO$_2$)/J(HONO). J(NO$_2$)/J(HONO) kept relatively constant (5.34~5.69) in the daytime in four seasons. Therefore, diurnal variation of [HONO] simulated by Eq. (10) depended on [NO$_2$] (Figure 1). This formula is only suitable for regions where the diurnal variation of [NO$_2$] is consistent with that of [HONO]. The parameterization described in this work was more reasonable and can be better used in the future in such coastal sites. Whether it is applicable to other types of sites needs to be further verified in the future. Figure 1 has been added in "Supplementary Material" named as Figure S5.*

[Figure]

**Figure 1. Diurnal variation of NO$_2$ concentration and HONO concentration simulated by Eq. (10)**

**Section 3.6, Page 14 – 15:** I think the authors should make a clear statement that they are evaluation the primary production of OH radical. As shown in many publications investigating HOx budgets, the production of OH during daytime is mainly by HO$_2$ / RO$_2$ + NO reaction.

*Response: Thanks for your valuable suggestion. The reaction of RO$_2$/HO$_2$ with NO is indeed an important source of OH radicals(Wang et al., 2018).Besides, O$_3$ photolysis, HONO photolysis, hydrogen peroxide photolysis and the ozonolysis of alkenes are main source of OH radicals. It is a pity that we do not have the supporting measurements to do this accurately except for O$_3$ and HONO. Therefore, we added a clear statement about the primary productions of OH radical in the manuscript like this: In addition to the two primary production of OH radicals mentioned above, there are the reaction of organic and hydro peroxy radicals (RO$_2$ and HO$_2$) with NO, O$_3$ photolysis, HONO photolysis, hydrogen peroxide photolysis and the ozonolysis of alkenes (Hofzumahaus et al., 2009;Gligorovski et al., 2015;Wang et al., 2018).*

**References**

Gligorovski, S., Strekowski, R., Barbati, S., and Vione, D.: Environmental Implications of Hydroxyl Radicals ((*)OH), Chem Rev, 115, 13051-13092, 10.1021/cr500310b, 2015.

Hofzumahaus, A., Rohrer, F., Lu, K., Bohn, B., Brauers, T., Chang, C.-C., Fuchs, H., Holland, F., Kita, K., Kondo, Y., Li, X., Lou, S., Shao, M., Zeng, L., Wahner, A., and Zhang, Y.: Amplified Trace Gas Removal in the Troposphere, Science, 324, 1702-1704, 2009.

Hou, S., Tong, S., Ge, M., and An, J.: Comparison of atmospheric nitrous acid during severe haze and clean periods in Beijing, China, Atmos. Environ. , 124, 199-206, 10.1016/j.atmosenv.2015.06.023, 2016.

Huang, R. J., Yang, L., Cao, J., Wang, Q., Tie, X., Ho, K. F., Shen, Z., Zhang, R., Li, G., Zhu, C., Zhang, N., Dai, W., Zhou, J., Liu, S., Chen, Y., Chen, J., and O'Dowd, C. D.: Concentration and sources of atmospheric nitrous acid (HONO) at an urban site in Western China, Sci. Total Environ., 593-594, 165-172, 10.1016/j.scitotenv.2017.02.166, 2017.

Kasibhatla, P., Sherwen, T., Evans, M. J., Carpenter, L. J., Reed, C., Alexander, B., Chen, Q., Sulprizio, M. P., Lee, J. D., Read, K. A., Bloss, W., Crilley, L. R., Keene, W. C., Pszenny, A. A. P., and Hodzic, A.: Global impact of nitrate photolysis in sea-salt aerosol on NOx, OH, and $O_3$ in the marine boundary layer, Atmos. Chem. Phys., 18, 11185-11203, 10.5194/acp-18-11185-2018, 2018.

Li, D., Xue, L., Wen, L., Wang, X., Chen, T., Mellouki, A., Chen, J., and Wang, W.: Characteristics and sources of nitrous acid in an urban atmosphere of northern China: Results from 1-yr continuous observations, Atmos. Environ., 182, 296-306, 10.1016/j.atmosenv.2018.03.033, 2018.

Li, X., Brauers, T., Häseler, R., Bohn, B., Fuchs, H., Hofzumahaus, A., Holland, F., Lou, S., Lu, K. D., Rohrer, F., Hu, M., Zeng, L. M., Zhang, Y. H., Garland, R. M., Su, H., Nowak, A., Wiedensohler, A., Takegawa, N., Shao, M., and Wahner, A.: Exploring the atmospheric chemistry of nitrous acid (HONO) at a rural site in Southern China, Atmos. Chem. Phys., 12, 1497-1513, 10.5194/acp-12-1497-2012, 2012.

Liu, Y., Nie, W., Xu, Z., Wang, T., Wang, R., Li, Y., Wang, L., Chi, X., and Ding, A.: Semi-quantitative understanding of source contribution to nitrous acid (HONO) based on 1 year of continuous observation at the SORPES station in eastern China, Atmos. Chem. Phys., 19, 13289-13308, 10.5194/acp-19-13289-2019, 2019.

Makkonen, U., Virkkula, A., M¨antykentt¨a, J., Hakola, H., Keronen, P., Vakkari, V., and Aalto, P. P.: Semi-continuous gas and inorganic aerosol measurements at a Finnish urban site: comparisons with filters, nitrogen in aerosol and gas phases, and aerosol acidity, Atmos. Chem. Phys., 12, 5617–5631, 10.5194/acp-12-5617-2012, 2012.

Meusel, H., Kuhn, U., Reiffs, A., Mallik, C., Harder, H., Martinez, M., Schuladen, J., Bohn, B., Parchatka, U., Crowley, J. N., Fischer, H., Tomsche, L., Novelli, A., Hoffmann, T., Janssen, R. H. H., Hartogensis, O., Pikridas, M., Vrekoussis, M., Bourtsoukidis, E., Weber, B., Lelieveld, J., Williams, J., Poschl, U., Cheng, Y. F., and Su, H.: Daytime formation of nitrous acid at a coastal remote site in Cyprus indicating a common ground source of atmospheric HONO and NO, Atmos. Chem. Phys., 16, 14475-14493, 10.5194/acp-16-14475-2016, 2016.

Nie, W., Ding, A. J., Xie, Y. N., Xu, Z., Mao, H., Kerminen, V.-M., Zheng, L. F., Qi, X. M., Huang, X., Yang, X.-Q., Sun, J. N., Herrmann, E., Petäjä, T., Kulmala, M., and Fu, C. B.: Influence of biomass burning plumes on HONO chemistry in eastern China, Atmos. Chem. Phys., 15 1147–1159, 10.5194/acp-15-1147-2015, 2015.

Park, S. S., Hong, S. B., Jung, Y. G., and Lee, J. H.: Measurements of $PM_{10}$ aerosol and gas-phase nitrous acid during fall season in a semi-urban atmosphere, Atmos. Environ., 38, 293-304, 10.1016/j.atmosenv.2003.09.041, 2004.

Qin, M., Xie, P., Su, H., Gu, J., Peng, F., Li, S., Zeng, L., Liu, J., Liu, W., and Zhang, Y.: An observational study of the HONO–$NO_2$ coupling at an urban site in Guangzhou City, South China, Atmos. Environ., 43, 5731-5742,

10.1016/j.atmosenv.2009.08.017, 2009.

Shi, X., Ge, Y., Zheng, J., Ma, Y., Ren, X., and Zhang, Y.: Budget of nitrous acid and its impacts on atmospheric oxidative capacity at an urban site in the central Yangtze River Delta region of China, Atmosp. Environ., 238, 117725, 10.1016/j.atmosenv.2020.117725, 2020.

Slanina, J., ten Brink, H. M., Otjes, R. P., Even, A., Jongejan, P., Khlystov, A., WaijersIjpelaan, A., and Hu, M.: The continuous analysis of nitrate and ammonium in aerosols by the steam jet aerosol collector (SJAC): extension and validation of the methodology, Atmos. Environ., 35, 2319-2330, 2001.

Spataro, F., Ianniello, A., Esposito, G., Allegrini, I., Zhu, T., and Hu, M.: Occurrence of atmospheric nitrous acid in the urban area of Beijing (China), Sci Total Environ, 447, 210-224, 10.1016/j.scitotenv.2012.12.065, 2013.

Stutz, J., Alicke, B., Ackermann, R., Geyer, A., Wang, S., White, A. B., Williams, E. J., Spicer, C. W., and Fast, J. D.: Relative humidity dependence of HONO chemistry in urban areas, J. Geophys. Res. Atmos., 109, n/a-n/a, 10.1029/2003jd004135, 2004.

Wang, H., Lyu, X., Guo, H., Wang, Y., Zou, S., Ling, Z., Wang, X., Jiang, F., Zeren, Y., Pan, W., Huang, X., and Shen, J.: Ozone pollution around a coastal region of South China Sea: interaction between marine and continental air, Atmos. Chem. Phys., 18, 4277–4295, 10.5194/acp-18-4277-2018, 2018.

Wang, J., Zhang, X., Guo, J., Wang, Z., and Zhang, M.: Observation of nitrous acid (HONO) in Beijing, China: Seasonal variation, nocturnal formation and daytime budget, Sci. Total Environ., 587-588, 350-359, 10.1016/j.scitotenv.2017.02.159, 2017.

Wang, S., Zhou, R., Zhao, H., Wang, Z., Chen, L., and Zhou, B.: Long-term observation of atmospheric nitrous acid (HONO) and its implication to local $NO_2$ levels in Shanghai, China, Atmos. Environ., 77, 718-724, 10.1016/j.atmosenv.2013.05.071, 2013.

Wen, L., Chen, T., Zheng, P., Wu, L., Wang, X., Mellouki, A., Xue, L., and Wang, W.: Nitrous acid in marine boundary layer over eastern Bohai Sea, China: Characteristics, sources, and implications, Sci. Total Environ., 10.1016/j.scitotenv.2019.03.225, 2019.

Wyers, G. P., Oties, R. P., and Slanina, J.: A continuous-flow denuder for the measurement of ambient concentrations and surface-exchange fluxes of ammonia. , Atmos. Environ., 27, 2085-2090, 1993.

Xu, Z., Wang, T., Wu, J., Xue, L., Chan, J., Zha, Q., Zhou, S., Louie, P. K. K., and Luk, C. W. Y.: Nitrous acid (HONO) in a polluted subtropical atmosphere: Seasonal variability, direct vehicle emissions and heterogeneous production at ground surface, Atmos. Environ., 10.1016/j.atmosenv.2015.01.061, 2015.

Ye, C., Zhou, X., Pu, D., Stutz, J., Festa, J., Spolaor, M., Tsai, C., Cantrell, C., Mauldin, R. L., 3rd, Campos, T., Weinheimer, A., Hornbrook, R. S., Apel, E. C., Guenther, A., Kaser, L., Yuan, B., Karl, T., Haggerty, J., Hall, S., Ullmann, K., Smith, J. N., Ortega, J., and Knote, C.: Rapid cycling of reactive nitrogen in the marine boundary layer, Nature, 532, 489-491, 10.1038/nature17195, 2016.

Yu, Y., Galle, B., Panday, A., Hodson, E., Prinn, R., and Wang, S.: Observations of high rates of $NO_2$-HONO conversion in the nocturnal atmospheric boundary layer in Kathmandu, Nepal, Atmos. Chem. Phys., 9 6401–6415, 2009.

Zheng, J., Shi, X., Ma, Y., Ren, X., Jabbour, H., Diao, Y., Wang, W., Ge, Y., Zhang, Y., and Zhu, W.: Contribution of nitrous acid to the atmospheric oxidation capacity in an industrial zone in the Yangtze River Delta region of China, Atmos. Chem. Phys., 20, 5457-5475, 10.5194/acp-20-5457-2020, 2020.

---

## Author Comment (AC2) · 3 Dec 2020

**AUTHOR RESPONSES IN BLUE ITALIC TEXT**

The manuscript "Exploration of the atmospheric chemistry of nitrous acid in a coastal city of southeastern China: Results from measurements across four seasons" by Hu et al. provides observations and analysis of compounds important for improving understanding of tropospheric chemistry. The topic is important to many readers, and this study is closely related to a large number of papers that try to understand atmospheric HONO abundance and its impact on oxidants. The writing is clear, and the observations are sufficiently unique and comprehensive to provide new insights. Regrettably, the utility of the manuscript is compromised by the omission of many important experimental details, such that the context and relevance of the measurements reported here cannot be ascertained. Additionally, there are several analyses that are incomplete or difficult to understand. Consequently, I cannot recommend this paper for publication. I do hope that these measurements will receive further examination and that a paper will be written that considers some of the suggestions below.

Response: Thanks for your positive feedback for the whole manuscript and valuable comments for some details. We have tried our best to improve the quality of this manuscript. Many experimental details have been added in the corresponding section to ascertain the context and relevance of the measurements. Several analyses have been improved to be complete and easy to understand.

The methodology section is far too brief, and many critical details are absent. The reference (Duan et al, 2018) that describes the HONO instrument notes the importance of characterizing HONO transmission and production in inlets. There is no mention of any of the sampling inlets. NO2 readily converts to HONO on surfaces, but there is no way to assess the importance of this artifact without a thorough description of inlet length, material, flow, etc. Are filters used on the IBBCEAS to remove ambient aerosol, as in Min et al., 2016? If so, how often are they changed? What are the uncertainties for the aerosol and NO measurements? It would be helpful to show how the IBBCEAS and TEI NO2 measurements compare. Line 105 says the TEI "might actually include other active nitrogen compounds". Did it? This assertion should be tested, or at least referenced.

Response: Thanks for your careful and precise working. The methodology section has been improved a lot as follows: The atmospheric concentrations of both HONO and NO2 were determined using IBBCEAS, which has previously been widely applied to such measurements (Tang et al., 2019;Duan et al., 2018;Min et al., 2016). The custom-built IBBCEAS instrument from the Anhui Institute of Optics and Fine Mechanic (AIOFM), Chinese Academy of Sciences, has been described in detail in previous study (Duan et al., 2018). Therefore, only a brief description is given here. Light was emitted by a single light-emitting diode (LED) with peak wavelength of 365 nm, full width at half maximum (FWHM) of 13 nm and was introduced into the resonant cavity, consisting of a pair of high-reflective (HR) mirrors with reflectivity of about 0.99983 at 368.2 nm, separated by 70 cm. The surface of the mirrors was purged by dry nitrogen at 0.1 Standard Liter per Minute (SLM), and the air flow was controlled by mass flow controller to prevent the surface of the mirror from being contaminated. The light transmitted through the cavity was received by an QE65000 spectrometer (Ocean Optics) through an optical fiber with 600 µm diameter and a 0.22 numerical aperture.

In order to avoid the drift of the center wavelength of the LED, the temperature of the LED was controlled to be approximately  $25\pm0.01$  °C by using a thermoelectric cooler unit. In order to prevent particulate matter from entering the cavity and reducing the effect of particulate matter on the effective absorption path, a 1 µm polytetrafluoroethylene (PTFE) filter membrane (Tisch Scientific) was used in the front end of the sampling port. In order to assure the quality of the data, the 1 µm PTFE filter membrane was usually replaced once every three days and the sampling tube was thoroughly cleaned with alcohol once a month. We increased the replacement frequency of the filter membrane and the cleaning frequency of the sampling tube in the event of heavy pollution to ensure that the filter membrane and sampling tube were in a clean state. The length of sampling tube with 6 mm outer diameter was approximately 3 m, the material was PFA with excellent chemical inertness and the sampling flow rate was 6 SLM meaning that the residence time of the gas in the sampling tube was less than 0.5 s. Besides, the sampling loss was calibrated before the experiment. We assessed the measured spectrum every day to ensure the authenticity of the measurement results. Multiple reflections in the resonator cavity enhanced the length of the effective absorption path, thereby enhancing the detection sensitivity of the instrument. The  $1\sigma$ detection limits for HONO and NO2 were about 60 ppt and 100 ppt, respectively, and the time resolution was 1 min. The fitting wavelength range was selected as 359–387 nm. Sample loss and secondary formation of HONO were both considered in this instrument and the measurement error of HONO was estimated to be approximately 9%. The sampling tube was heated to 35 °C and covered by insulation cotton materials to prevent the effect of condensation of the water vapor(Lee et al., 2013).

The uncertainties for the aerosol and NO measurements were10-20%(Tian et al., 2016) and 10%(Xu et al., 2015), respectively. As shown in Figure 1, the NO2 concentration measured by IBBCEAS had the same trend as the NO2 concentration measured by TEI 42i, and NO2 concentration measured by IBBCEAS is always lower than that by TEI 42i. The average NO2 concentration determined by IBBCEAS and TEI 42i were 14.99 ppb and 18.68 ppb, respectively. Besides, this result also proved by (Villena et al., 2012;Xu et al., 2019;Zheng et al., 2020) that chemiluminescence instruments used for indirect NO2 detection in monitoring networks were affected by other active nitrogen components. The manuscript has been revised as follows: As expected, the NO2 concentration measured by IBBCEAS had the same trend as the NO2 measured by TEI 42i, and NO2 concentration measured by IBBCEAS had the same trend as the NO2 measured by TEI 42i, and NO2 concentration measured by IBBCEAS had the same trend as the NO2 measured by TEI 42i, and NO2 concentration measured by IBBCEAS had the same trend as the NO2 measured by TEI 42i, and NO2 concentration measured by IBBCEAS had the same trend as the NO2 measured by TEI 42i, and NO2 concentration measured by IBBCEAS had the same trend as the NO2 measured by TEI 42i, and NO2 concentration measured by IBBCEAS had the same trend as the NO2 measured by TEI 42i, and NO2 concentration measured by IBBCEAS had the same trend as the NO2 measured by TEI 42i, and NO2 concentration measured by IBBCEAS had the same trend as the NO2 measured by TEI 42i, and NO2 concentration measured by IBBCEAS had the same trend as the NO2 measured by TEI 42i, and NO2 concentration measured by IBBCEAS had the same trend as the NO2 measured by TEI 42i, and NO2 concentration measured by IBBCEAS had the same trend as the NO2 measured by TEI 42i, and NO2 concentration measured by IBBCEAS had the same trend as the NO2 measured by TEI 42i.